# Liver-Targeted Scutellarin Nanoemulsion Alleviates Fibrosis with Ancillary Modulation of the Gut–Liver Microbiota

**DOI:** 10.3390/ijms26199746

**Published:** 2025-10-07

**Authors:** Haoyang Yu, Xia Niu, Bingyu Niu, Peng Lei, Ning Xu, Sitong Yang, Quanyong Yu, Guiling Li, Lulu Wang

**Affiliations:** 1State Key Laboratory of Bioactive Substance and Function of Natural Medicines, Institute of Medicinal Biotechnology, Chinese Academy of Medical Sciences & Peking Union Medical College, Beijing 100050, China; yuhy1014@163.com (H.Y.); niuxia307@163.com (X.N.); balala482@163.com (B.N.); xuning0624@163.com (N.X.); yangst0323@163.com (S.Y.); 2Institute of Materia Medica, Chinese Academy of Medical Sciences & Peking Union Medical College, Beijing 100050, China; leipeng156@163.com; 3School of Pharmacy, Jiangning Campus, China Pharmaceutical University, Nanjing 210009, China; yuquanyong04@163.com

**Keywords:** liver fibrosis, scutellarin, gut microbiota, hepatic microbiota, liver-targeted delivery

## Abstract

Liver fibrosis, a progressive condition with limited pharmacotherapies, poses a global health challenge. Scutellarin (SCU), a flavonoid derived from *Erigeron breviscapus*, has demonstrated anti-fibrotic activity and modulates gut microbiota. Emerging evidence suggests that SCU may also influence the hepatic microbiome. However, its clinical utility is constrained by poor water solubility and low oral bioavailability. Here, we developed an SCU-loaded nanoemulsion (SCE) to enhance solubility and liver-targeted delivery. In vitro, SCE increased SCU uptake in hepatic stellate cells (HSCs) and significantly inhibited TGF-β1-induced fibrogenesis. In a bile duct ligation (BDL) mouse model, oral administration of SCE improved hepatic SCU accumulation and produced superior anti-fibrotic efficacy. SCE treatment attenuated fibrosis and collagen deposition in the liver and improved liver function markers. Mechanistic investigations using 16S rRNA sequencing revealed that SCU treatment was associated with beneficial microbiota changes, although its main therapeutic effects were achieved through enhanced hepatic targeting. Notably, the SCE formulation was well-tolerated, showing no significant toxicity in vitro or in vivo. In conclusion, the SCU-loaded nanoemulsion achieved enhanced hepatic delivery of SCU and exerted potent anti-fibrotic effects via multiple mechanisms, including direct suppression of fibrogenesis and ancillary modulation of the gut–liver microbiome, offering a promising therapeutic strategy for liver fibrosis.

## 1. Introduction

Liver fibrosis is a chronic condition characterized by the excessive accumulation of extracellular matrix proteins, primarily due to the differentiation of hepatic stellate cells (HSCs) into myofibroblasts during persistent liver injury. This pathological process disrupts the normal hepatic architecture and function, ultimately leading to cirrhosis and liver failure if untreated [1]. Managing liver fibrosis remains a major clinical challenge because effective anti-fibrotic therapies are scarce. Current treatments mainly address the underlying cause of injury—for example, antiviral therapy for hepatitis or alcohol abstinence in alcoholic liver disease—but they achieve only limited reversal of established fibrosis [2]. To date, no universally approved anti-fibrotic drug exists for broad clinical use. For instance, Rezdiffra (resmetirom) has demonstrated therapeutic benefit in patients with noncirrhotic NASH (nonalcoholic steatohepatitis) and fibrosis [3]; however, its efficacy across other etiologies of liver fibrosis remains unproven. These limitations underscore the urgent need for new therapeutic strategies that directly target established fibrosis, irrespective of etiology.

The gut–liver axis has been increasingly recognized as a critical determinant of liver health and disease. The gut microbiota, through its metabolic activities and cellular components, can influence the progression of liver fibrosis [4]. Dysbiosis—an imbalance in the gut microbial community—can increase intestinal permeability, allowing bacterial products such as endotoxins to translocate into the portal circulation. These products trigger hepatic inflammation and exacerbate fibrogenesis. Consequently, modulating the gut microbiota has emerged as a promising therapeutic strategy for liver disease. Previous studies have shown that scutellarin (SCU), a natural flavonoid derived from *Erigeron breviscapus* (Compositae), regulates intestinal microbiota composition [5,6]. This finding suggests that SCU’s anti-fibrotic activity may partly depend on its ability to improve gut microbial balance. In line with the evolving paradigm of multi-target therapies for complex diseases [7], we hypothesized that SCU might also influence the liver’s local microbiome. Emerging evidence, including our recent work, supports the existence of a resident “hepatic microbiota” that may contribute to liver fibrosis progression [8]. Thus, SCU’s therapeutic effects may extend to modulating hepatic microbiota and, in turn, fibrogenesis. Nonetheless, research on the liver microbiome remains at an early stage, and further studies are needed to identify microbial changes that are beneficial or harmful in liver fibrosis.

SCU, a well-studied bioactive ingredient in traditional Chinese medicine, exhibits a broad spectrum of pharmacological activities. These include cardioprotective effects, such as the attenuation of cardiac hypertrophy [9] and the amelioration of ischemia–reperfusion injury [10,11], as well as antimicrobial [12] and antiviral properties [13]. SCU also exhibits neuroprotective effects, for example, in glaucoma [14], and antitumor activities through modulation of immune responses [15]. Notably, SCU has demonstrated anti-fibrotic efficacy in other organ systems, including the amelioration of cardiac interstitial fibrosis post-infarction [16] and the attenuation of pulmonary fibrosis [17]. Collectively, these findings suggest that SCU has potential as an anti-fibrotic agent in hepatic disease. Given its diverse pharmacological profile and anti-fibrotic activity in the heart and lung, it is reasonable to propose that SCU could also play a beneficial role in ameliorating liver fibrosis, especially if delivered effectively to hepatic targets.

Despite its therapeutic potential, the clinical use of SCU is limited by its unfavorable biopharmaceutical properties. Specifically, SCU’s poor water solubility markedly limits its oral absorption and bioavailability, thereby reducing its efficacy. Overcoming these challenges requires advanced drug delivery strategies that improve solubility, protect SCU from degradation during gastrointestinal transit, and enable targeted delivery to the liver. Recent innovations, including phospholipid complexation and nanoformulations, have enhanced the solubility and bioavailability of poorly water-soluble drugs and, in some cases, conferred organ-targeting capabilities [18].

In this study, we developed a novel nanoemulsion-based delivery system for SCU to improve its solubility and liver-targeting capability. Initially, we prepared a scutellarin–phospholipid complex (SPC) to increase the lipophilicity of SCU and facilitate its incorporation into the nanoemulsion. Using this complex, we formulated a scutellarin-loaded nanoemulsion (SCE) stabilized with chitosan oligosaccharide. We then conducted a comprehensive physicochemical characterization of both SPC and SCE and evaluated their performance in vitro and in vivo. Specifically, we examined the ability of SCE to enhance cellular uptake and hepatic accumulation of SCU, along with its pharmacokinetic and pharmacodynamic profiles in a bile duct ligation (BDL)-induced liver fibrosis mouse model. We further assessed the therapeutic efficacy of SCE against liver fibrosis in vivo and in vitro, using a TGF-β1-activated HSC (LX-2 cell) model. In addition, we investigated whether free and nanoformulated SCU could modulate gut and liver microbiota in fibrotic mice, exploring the gut–liver axis as a potential mechanism of action. Finally, we confirmed the safety and biocompatibility of SCE through both in vitro assays (normal liver cells and LX-2 cells) and in vivo assessments (histopathology and blood chemistry in mice). Overall, our findings demonstrate that SCE significantly alleviates liver fibrosis by enhancing hepatic delivery and multi-target mechanisms, offering a promising therapeutic approach that combines microbiome modulation with direct anti-fibrotic activity.

## 2. Results

### 2.1. SCU Modulates Gut and Liver Microbiota Composition in BDL-Induced Fibrosis

#### 2.1.1. Gut Microbiota

We first examined the impact of SCU on gut microbiota dysbiosis caused by BDL-induced liver fibrosis. High-throughput 16S rRNA sequencing of fecal samples yielded 2,164,399 high-quality sequences (average > 240,000 reads per sample), which clustered into 2394 OTUs (at 97% similarity) at 97% similarity across all groups. Sequencing depth was sufficient to capture the majority of microbial diversity, as indicated by the plateauing rarefaction curves (Appendix A). Alpha diversity indices, including Chao1 richness and ACE, revealed no significant differences in overall bacterial richness across the Sham, BDL, and SCU-treated groups at the phylum, genus, or OTU levels (Appendix A). However, beta diversity analysis revealed substantial alterations in microbial community structure in the BDL group. PCoA and PLS-DA demonstrated distinct clustering of the BDL group’s microbiota composition compared to the Sham controls, indicating fibrosis-induced changes in gut microbial composition (Figure 1A).

Although overall diversity did not change significantly, BDL induced notable taxonomic alterations in the gut microbiota. At the phylum level, BDL slightly increased the relative abundance of several potentially harmful bacterial phyla, including *Actinobacteriota*, *Cyanobacteria*, and *Desulfobacterota*, compared to the Sham group (Figure 1B,C). These changes, while subtle, are consistent with disease-associated dysbiosis reported in the literature. For instance, elevated *Actinobacteriota* abundance has been associated with steatosis severity in mouse models [19] and has been observed in both gut and liver microbiota in hepatocellular carcinoma [20]. Similarly, the elevated abundance of *Desulfobacterota*—a group comprising sulfate-reducing bacteria—has been linked to hepatic inflammation and injury in other studies [21], while overgrowth of *Cyanobacteria* and *Desulfobacterota* has been implicated in gut dysbiosis and various diseases [22,23,24,25]. At the family level, BDL increased the abundance of *Staphylococcaceae* and *Desulfovibrionaceae* (Appendix A), both of which include pathogenic members. Specifically, increased abundance of *Staphylococcaceae* has been reported in pediatric nonalcoholic fatty liver disease [26] and cirrhosis-associated gut microbiota profiles [27], whereas *Desulfovibrionaceae* (sulfate-reducing, endotoxin-producing bacteria) have been implicated in steatohepatitis and liver cancer [28,29]. Conversely, BDL suppressed the abundance of beneficial taxa such as the genus *Clostridia_UCG-014* (family *Clostridiaceae*), which includes butyrate-producing bacteria linked to improved metabolic and inflammatory profiles [30]. BDL also increased the abundance of several genera considered harmful or pro-inflammatory, including *Bacteroides*, *Dubosiella*, and *Lacticigenium*. Notably, *Bacteroides* spp. are known to overgrow in gut dysbiosis and can produce metabolites that contribute to inflammation [31,32]. Similarly, increased *Dubosiella* abundance (family *Erysipelotrichaceae*) has been reported in certain inflammatory conditions [33], whereas *Lacticigenium* (a lactic acid bacterium) has been linked to gut metabolic disturbances [34].

Importantly, SCU treatment corrected many of these BDL-induced microbial changes. Although some differences did not reach statistical significance due to variability, the microbiota composition of SCU-treated mice trended toward the Sham group profile. SCU administration reduced the relative abundance of *Actinobacteriota*, *Cyanobacteria*, and *Desulfobacterota* compared to untreated BDL mice (Figure 1C). At the family level, SCU reduced the abundance of *Staphylococcaceae* and *Desulfovibrionaceae*, while partially restoring *Clostridiaceae* populations, like *Clostridia_UCG-014* (Appendix A). Similarly, the overgrowth of *Bacteroides*, *Dubosiella*, and *Lacticigenium* observed in BDL mice was moderated in the SCU group. These results suggest that SCU can beneficially modulate the gut microbiota in liver fibrosis, reducing potentially deleterious bacteria and supporting the growth of beneficial taxa. Overall, SCU appears to mitigate the dysbiosis induced by BDL, shifting the microbial community toward a healthier composition.

#### 2.1.2. Liver Microbiota

We subsequently examined the liver microbiome composition in the same cohort of mice. Although liver tissue contains low microbial biomass, 16S rRNA gene sequencing was successful, yielding 1,491,466 high-quality sequences, with an average of approximately 165,000 reads per sample. Clustering at 97% sequence similarity identified 8512 OTUs. As expected, liver tissue exhibited a broader diversity of taxa than fecal samples, likely reflecting blood-borne or translocated bacteria. In total, sequences were classified into 34 phyla, 436 families, and 891 genera. Rarefaction analysis confirmed sufficient sequencing depth (Appendix A). Alpha diversity analyses revealed subtle differences among groups (Appendix A). BDL decreased the liver microbial richness and diversity compared to Sham, whereas SCU treatment appeared to prevent this decrease, although the changes in ACE and Chao1 indices were not statistically significant. Beta diversity, assessed by PLS-DA, revealed distinct clustering among the three experimental groups, indicating that both fibrotic injury and SCU intervention impacted liver microbial composition (Figure 2A).

Taxonomic comparisons revealed several notable changes in liver microbiota composition due to BDL and its modulation by SCU. At the order level, the relative abundance of *Erysipelotrichales* was significantly elevated in BDL mice compared to Sham controls (Figure 2B,C), consistent with previous studies linking this order to high-fat diet-induced nonalcoholic steatohepatitis (NASH) and disease progression [35]. Notably, SCU administration significantly reduced the abundance of *Erysipelotrichales*, restoring levels close to those of the Sham group (Figure 2C), suggesting a corrective effect on fibrosis-associated microbial dysregulation. Additionally, SCU had a mild regulatory effect on other taxa perturbed by BDL. At the genus level, *Oscillibacter* (family *Ruminococcaceae*) emerged as another taxon affected by BDL. Its abundance was significantly reduced in the livers of BDL mice relative to Sham controls but was partially restored following SCU treatment (Figure 2E). Although the role of *Oscillibacter* in liver disease remains unclear, with some studies associating it with anti-inflammatory properties and others linking it to disease states [36,37], its partial normalization in response to SCU suggests a potentially beneficial modulation. In addition to these major changes, BDL induced several minor alterations in the abundance of families within the *Proteobacteria* and *Firmicutes* phyla, which were also moderated by SCU treatment. Collectively, these data indicate that SCU regulates the liver microbial community in fibrotic mice, paralleling its influence on the gut microbiota. Given the relatively limited research on the hepatic microbiome, these findings should be interpreted with caution. Nonetheless, they suggest that SCU can influence microbial populations in the liver. Given that the oral route initially exposes the drug to gut microbiota before systemic absorption, part of the therapeutic effect may arise from modulation of the gut–liver axis. However, the enhanced hepatic accumulation achieved by the nanoemulsion suggests that these microbial alterations are likely secondary and supportive to SCU’s primary liver-targeted antifibrotic action.

These findings suggest that SCU modulates the liver-resident microbiota disturbed by cholestatic injury, although the functional consequences of these changes remain unclear. The therapeutic benefits of SCE in BDL-induced fibrosis involve a multifaceted mechanism. Our pharmacokinetic and tissue distribution data confirm significantly enhanced hepatic accumulation of SCU, which correlates with the observed reduction in the expression of liver fibrosis markers, suggesting direct pharmacodynamic action at the target site. Simultaneously, oral administration exposes SCU transiently to the gut microbiota, partially correcting fibrosis-associated dysbiosis. Although microbial modulation may be limited by the nanoemulsion’s efficient hepatic uptake, this early-phase interaction with the gut epithelium and microbiota could influence immune-metabolic pathways via the gut-liver axis. While these microbiota-mediated effects likely do not constitute the primary mechanism, they may complement and reinforce SCU’s antifibrotic action. Overall, this integrated perspective highlights the dual advantage of oral nanoformulations: targeted hepatic delivery coupled with supportive modulation of gut microbiota. Given the relatively low bacterial biomass in liver tissue, the results from liver 16S sequencing should be interpreted as exploratory with caution.

### 2.2. Preparation and Physicochemical Characterization of SPC and SCE

#### 2.2.1. SCU–Phospholipid Complex (SPC) Formation

SPC was successfully prepared using the solvent evaporation method, and its formation was confirmed through spectroscopic and thermal analyses. The FTIR spectra (Figure 3C) revealed that pure SCU has characteristic O–H stretching vibration peaks around 3500–3300 cm^−1^ due to phenolic hydroxyl groups and distinct absorption bands in the 1500–1700 cm^−1^ region due to aromatic ring vibrations and carbonyl groups. In the physical mixture of SCU and phospholipid, peaks from both components were present without significant shifts, indicating just a simple superposition of spectra. However, the FTIR spectrum of SPC exhibited notable differences: the broad O–H stretch band was greatly diminished or shifted, and several peaks in the 1500–1700 cm^−1^ region were attenuated or disappeared. These changes suggest strong intermolecular interactions in SPC, likely due to hydrogen bonding between SCU and phospholipid molecules, which can alter the vibrational frequencies of functional groups.

Thermal analysis by DSC provided further evidence of complex formation. The thermogram of pure SCU displayed endothermic peaks at 163.6 °C and 353.3 °C, which are attributed to melting or decomposition processes of SCU, as well as an exothermic peak at 206.7 °C, likely due to a phase transition or crystallization event. These thermal transitions confirm the crystalline nature of SCU. The physical mixture exhibited endothermic peaks around 234 °C, corresponding to the melting of the phospholipid (which has a broad transition) and perhaps a shifted SCU melt due to some mixing. In contrast, the DSC thermogram of SPC (Figure 3D) showed no sharp endothermic or exothermic peaks over the range up to 400 °C, indicating the absence of the characteristic thermal transitions associated with crystalline SCU. This absence suggests that SCU exists in an amorphous or molecularly dispersed state within the phospholipid matrix. The phospholipid likely stabilizes SCU in this amorphous state, which is typically associated with enhanced solubility.

XRD analysis further substantiated the structural transformation of SCU in the complex. The XRD patterns (Figure 3E) of pure SCU displayed numerous distinct peaks, confirming its crystalline nature. The phospholipid showed a few broad peaks, while the physical mixture retained the distinct diffraction patterns of both components, suggesting no significant interaction upon simple grinding. Notably, SPC exhibited a diffractogram with no pronounced diffraction peaks. Instead, only a diffuse halo was observed, characteristic of amorphous substances. The disappearance of SCU’s crystalline peaks in SPC confirms that SCU was successfully converted into a non-crystalline form through complexation with the phospholipid. Together, the FTIR, DSC, and XRD results demonstrate that SCU and the phospholipid formed a complex, likely via hydrogen bonding and hydrophobic interactions, resulting in an amorphous solid. This complexation is expected to enhance the apparent solubility of SCU in the oil phase, providing a basis for the subsequent nanoemulsion formulation.

#### 2.2.2. Nanoemulsion Properties and Stability

Incorporating SPC into the nanoemulsion produced a milky white formulation, which became transparent with a slight bluish opalescence upon sufficient dilution in water (Figure 3F). This Tyndall effect qualitatively indicates the presence of nanoscale particles. DLS confirmed that the SCE droplets were indeed in the nanometer range. The average particle size of the freshly prepared nanoemulsion was 78.5 ± 2.5 nm (Figure 3H), with a PDI of approximately 0.20, suggesting a relatively narrow size distribution. The zeta potential was slightly negative at −3.8 mV, likely due to the combined effects of the negatively charged phospholipid and the partially cationic COS coating. TEM imaging confirmed the droplet morphology and size, revealing roughly spherical and uniform particles ranging from 70 to 100 nm (Figure 3G), consistent with the DLS results.

The SCE formulation also exhibited satisfactory short-term stability at 4 °C. During a 7-day storage period, its characteristics changed minimally. The mean particle size increased slightly, from approximately 78 nm to nearly 89 nm by day 7, while the PDI remained between 0.19 and 0.23, indicating no significant aggregation or broadening of size distribution. The zeta potential stayed within −3 to −4 mV throughout the period. Notably, the SCU content in the nanoemulsion remained above 90% of the initial concentration, with fluctuations below 10% over the study period (Table 1). These results confirm that SCE maintains both physical and chemical stability under refrigerated conditions for at least one week, sufficient for experimental use and suggesting potential for practical shelf-life upon further optimization. Taken together, the characterization results show that the SPC-based nanoemulsion was successfully prepared, yielding nanosized SCU carriers with suitable stability. This short-term stability was sufficient for the experimental duration of this study. However, long-term stability over periods of months, which is critical for clinical translation, was not assessed and will be an important focus of future formulation optimization studies.

### 2.3. SCE Formulation Enhances Cellular Uptake of SCU

We next investigated whether nanoemulsion-based delivery could enhance SCU uptake by SCU by HSCs. To this end, LX-2 cells were employed to compare the cellular uptake kinetics of SCU delivered via SCE versus free SCU. Because SCU exhibits limited intrinsic fluorescence, we used NR, a lipophilic fluorescent probe, as a surrogate tracer. Confocal microscopy revealed pronounced differences in uptake dynamics between the two formulations. In cells treated with free NR (dissolved in PBS), virtually no red fluorescence was detectable in the cytoplasm during the first hour of incubation. A faint signal appeared by 2 h and became more apparent by 4 h, indicating slow diffusion of NR into cells (Figure 4B). In contrast, cells treated with NR-loaded SCE showed detectable red fluorescence as early as 5 min after exposure, and the intensity increased rapidly over time. At 15 min, 30 min, and 1 h, the NR signal in the SCE group was already substantial, whereas it remained negligible in the free NR group at those same time points. Even at later time points (2 h and 4 h), the fluorescence intensity in SCE-treated cells significantly exceeded that of cells treated with free NR. This qualitative observation implies that the nanoemulsion facilitated a much faster and higher uptake of the hydrophobic compound into HSCs.

Quantitative assessment by flow cytometry confirmed the confocal microscopy observations. The mean fluorescence intensity of NR in LX-2 cells treated with SCE was consistently and significantly higher than in cells treated with free NR at all measured time points (Figure 4C). For example, at 1 h, NR fluorescence in the SCE group was several-fold greater than in the free NR group, consistent with the confocal imaging results. These results collectively demonstrate that nanoemulsion-based delivery markedly enhances the intracellular uptake of hydrophobic compounds into HSCs, likely due to the efficient endocytosis of nano-sized droplets.

To elucidate the mechanisms underlying the enhanced cellular uptake of SCE, we employed a panel of endocytosis inhibitors. As shown in Figure 4D,E, caveolae-mediated endocytosis emerged as the primary pathway for SCE internalization. Specifically, treatment with genistein (a caveolin-dependent endocytosis inhibitor) or methyl-β-cyclodextrin (which depletes membrane cholesterol and disrupts lipid raft/caveolae structures) significantly reduced NR uptake from SCE, suggesting these pathways mediate the majority of the nanoemulsion’s uptake. Conversely, EIPA (an inhibitor of macropinocytosis) had no significant effect on SCE uptake, indicating macropinocytosis was not involved. Chlorpromazine, an inhibitor of clathrin-mediated endocytosis, caused only a slight decrease in uptake, implying that clathrin pathways play a secondary role. Similarly, nystatin, another caveolae pathway inhibitor, reduced uptake. These findings suggest that SCE enters LX-2 cells primarily through caveolae/caveolin-1-mediated endocytosis, a pathway commonly utilized by nanoparticles for efficient intracellular delivery. The nanoemulsion’s small size and composition may likely promote its association with lipid rafts and caveolar pits on the cell membrane, driving rapid internalization. In summary, formulating SCU into SCE significantly increases its cellular uptake by HSCs via energy-dependent, caveolae-mediated endocytosis, which is expected to improve intracellular bioavailability and therapeutic efficacy.

### 2.4. SCE Alleviates Liver Fibrosis In Vitro

Having established that SCE improves SCU uptake, we next assessed whether this improvement translates into greater anti-fibrotic efficacy in vitro. LX-2 cells activated with TGF-β1 were used as a model of fibrogenic HSCs. We evaluated two hallmark features of activated HSCs in the presence or absence of SCU treatments: increased migratory capacity and overexpression of fibrosis-related genes.

In the scratch wound healing assay, untreated activated LX-2 cells (model group) migrated rapidly to close the scratch, reflecting their high motility upon activation. After 12 h, the wound area in the model group was partly filled by migrating cells (Figure 5A). In contrast, LX-2 cells treated with free SCU exhibited a modest reduction in wound closure; the scratch area remained slightly more open at 12 h compared to the model group, suggesting that SCU can attenuate HSC migration to some extent. Notably, SCE-treated cells exhibited a substantially larger remaining scratch area at 12 h, demonstrating significant impairment in migration. By 24 h, the differences between groups were even more pronounced, indicating a strong anti-migratory effect of the nanoemulsion. These results indicate that while free SCU can suppress HSC migration to some degree, SCE substantially enhanced this activity, likely due to improved intracellular delivery of SCU.

We next examined fibrogenic gene expression in LX-2 cells. Real-time PCR analysis (Figure 5B) demonstrated that TGF-β1 stimulation markedly upregulated key fibrogenesis-associated genes such as *ACTA2* (encoding α-SMA) and *COL1A1* (collagen type I) in LX-2 cells (model vs. quiescent control). Treatment with free SCU produced only a modest reduction in gene expression, which did not reach statistical significance relative to the model group. In contrast, SCE treatment significantly suppressed both *ACTA2* and *COL1A1* expression. Specifically, α-SMA mRNA in SCE-treated cells decreased to nearly baseline levels, and collagen I mRNA was greatly reduced, indicating an effective suppression of HSC activation at the transcriptional level. These results suggest that the intracellular SCU delivered via SCE effectively interfered with TGF-β1–driven fibrogenic signaling.

Consistent with the mRNA data, Western blot analysis confirmed that SCE more effectively suppressed fibrotic protein production in LX-2 cells compared to free SCU (Figure 5C,D). Untreated activated HSCs exhibited high collagen I expression, as expected for myofibroblastic cells, and elevated MMP2 levels, reflecting matrix remodeling activity despite excessive collagen accumulation. Free SCU treatment marginally reduced *collagen I* and *MMP2* levels relative to the model group. In contrast, SCE markedly downregulated *ACTA2*, *COL1A1*, and *MMP2* expression, indicating a shift toward a less fibrogenic, more quiescent phenotype. Densitometric analysis indicated that *collagen I* and *MMP2* in the SCE group were decreased by nearly 40–50% compared to the model, whereas free SCU achieved only approximately 10–20% reductions. These findings demonstrate that SCU can mitigate HSC activation and fibrogenesis, and that delivering SCU via the nanoemulsion significantly amplifies these anti-fibrotic effects. By enhancing cellular uptake and retention of SCU in HSCs, SCE achieves better suppression of the fibrotic phenotype (reduced motility, collagen production, and myofibroblastic markers) than the free drug.

### 2.5. SCE Exhibits Enhanced Liver-Targeting In Vivo

We then evaluated whether the nanoemulsion formulation could improve the delivery of SCU to the liver in vivo, a key factor for maximizing therapeutic efficacy against liver fibrosis. To visualize and quantify tissue distribution, we used the near-infrared fluorescent dye DIR as a surrogate for SCU, incorporated either into the nanoemulsion or administered as a free compound. Following oral administration in mice, we monitored real-time in vivo fluorescence and performed ex vivo imaging of major organs to compare biodistribution profiles. In contrast, mice treated with DIR-loaded SCE showed a notably stronger fluorescence signal in the liver region at early time points, suggesting that more of the administered dose reached the liver, likely via uptake of nanoemulsion through intestinal lymphatics and subsequent accumulation in the liver.

Ex vivo fluorescence imaging of major organs revealed marked enhancement of liver targeting by SCE (Figure 6A). At 1 h post-treatment, livers from SCE-treated mice fluoresced more intensely than those from mice treated with free DIR. This difference became increasingly evident over time: by 6 h, the fluorescence signal in the free DIR group had declined, whereas SCE-treated livers retained high intensity. At 12 h, the liver signal in the free DIR group had largely dissipated, while SCE-treated livers remained brightly fluorescent. Even at 24 h, livers from the SCE group emitted a detectable signal, indicating prolonged retention of the nanoemulsion or its payload in hepatic tissue, whereas the free DIR was almost undetectable in the liver by that time. Quantitative analysis of liver fluorescence confirmed these observations (Figure 6B). The area-normalized radiant efficiency (or total photon count) in the livers of SCE-treated mice was significantly higher at all measured time points compared to the free DIR group. For instance, the peak liver fluorescence (observed around 3–6 h) in the SCE group was several-fold greater than the peak in the free group. Moreover, while the free DIR liver signal decreased by approximately 52% between 6 h and 24 h, the SCE liver signal only declined by nearly 8% over the same period, indicating that SCE significantly prolongs the residence time of the compound in the liver.

In addition to enhancing liver targeting, the nanoemulsion formulation reduced or delayed distribution to non-target organs. In the SCE group, fluorescence in organs other than the liver remained low beyond the first hour post-administration. In contrast, mice treated with free DIR exhibited a more diffuse fluorescence pattern, with substantial signal detected in the lungs and kidneys shortly after dosing, suggesting rapid systemic distribution and clearance. This difference is likely attributable to the distinct absorption pathway of the nanoemulsion. With a particle size of approximately 80 nm, SCE is efficiently taken up via Peyer’s patches and intestinal lymphatics, bypassing immediate hepatic (first-pass) metabolism as intact particles. The nanoemulsion then accumulates in the liver, either within the fenestrated endothelium or through uptake by Kupffer cells and hepatocytes. This pathway reduces plasma spikes and renal excretion, enabling sustained delivery of the payload to the liver over time.

In summary, the in vivo imaging results highlight the pronounced liver-targeting capacity of the nanoemulsion formulation. SCE delivers a greater proportion of SCU to the liver and prolongs its retention compared to an equivalent dose of free SCU. This enhanced hepatic delivery is expected to translate into improved anti-fibrotic efficacy, as more drug is available at the site of pathology for an extended period. Consistent with these findings, SCE treatment produced markedly superior therapeutic outcomes in the BDL model relative to free SCU, consistent with the observed improvements in pharmacokinetics.

### 2.6. Therapeutic Efficacy of SCE in BDL-Induced Liver Fibrosis

The ultimate goal of developing SCE was to achieve superior therapeutic effects against liver fibrosis. To evaluate this, we employed the BDL mouse model and compared pathological and biochemical outcomes among groups treated with SCE, free SCU, or vehicle. Sham-operated mice served as healthy controls.

Gross examination of the livers at sacrifice revealed visible signs of treatment efficacy. Livers from untreated BDL mice were enlarged, cholestatic (yellow-brown), and had an irregular surface with nodularity and bile accumulation, indicative of severe injury and fibrosis (Figure 7A). Livers from the free SCU group exhibited modest improvements, with reduced swelling and less discoloration; however, they remained morphologically abnormal. In contrast, livers from the SCE-treated group displayed near-normal morphology: they were smaller in size (less hepatomegaly), with a smoother surface more akin to Sham livers, and fewer visible lesions. This visual improvement suggested that SCE substantially mitigated the BDL-induced liver damage.

Histological analyses confirmed that SCE was more effective in reducing liver injury and fibrosis (Figure 7B–D). H&E staining of livers from the BDL group revealed extensive architectural disruption, with large areas of hepatocyte necrosis, inflammation, ductular proliferation, and fibrosis. Free SCU treatment led to a modest improvement: H&E sections from this group still exhibited significant damage, but with somewhat reduced necrotic areas and inflammation compared to untreated BDL. However, l SCE treatment markedly improved tissue architecture: necrotic foci were rare, inflammatory cell infiltration was reduced, and normal hepatocyte cords were more apparent. Although fibrotic septa remained, their extent was significantly diminished.

We performed Masson’s trichrome and Sirius Red staining to visualize collagen deposition. Untreated livers from the BDL group showed extensive bridging fibrosis, indicated by dense blue (Masson) and red (Sirius Red) staining between portal areas. Free SCU treatment slightly reduced the thickness and extent of fibrotic bands, suggesting a partial reduction in collagen deposition. Remarkably, SCE treatment markedly attenuated fibrosis: collagen-positive septa became thinner, more fragmented, and localized, while large collagen-free parenchymal regions resembled an earlier stage of fibrosis. Quantitative image analysis confirmed these findings (Figure 7E,F). SCE significantly reduced the collagen-positive area compared to both the model and free SCU groups (*p* < 0.01), whereas the free SCU group showed no significant difference from the untreated model group in Sirius Red quantification. Thus, SCE curtailed the progression of fibrosis, halting it at a much lower level of collagen deposition.

Biochemical analyses of liver function further corroborated the therapeutic benefit of SCE (Figure 7G–J). The BDL model group exhibited very high serum levels of ALT and AST, reflecting substantial hepatocellular injury, along with elevated ALP and total bile acids (TBA) due to cholestasis from bile duct ligation. Free SCU treatment modestly decreased the levels of these markers, but the reductions were not statistically significant. In free SCU-treated mice, these liver injury markers were slightly lower on average, but the differences were not statistically significant for most markers, indicating limited hepatoprotection by free SCU. In contrast, SCE significantly improved all parameters: it reduced ALT and AST by approximately 50% and decreased ALP and TBA levels similarly, reflecting both hepatoprotection and enhanced bile flow. Across all groups, SCE restored liver function closest to Sham levels—for example, ALT in SCE-treated mice remained only slightly above Sham, whereas BDL mice exhibited several-fold higher levels. Combined with the histological findings, these results demonstrate that SCE provided the greatest therapeutic benefit, effectively protecting hepatocytes, reducing fibrosis, and improving overall liver function, far outperforming the free SCU treatment.

To further validate the anti-fibrotic efficacy of SCE, we analyzed molecular markers of fibrosis and inflammation in the liver tissues. Western blot results (Figure 8A–C) showed that BDL strongly upregulated α-SMA, collagen I, and the pro-inflammatory cytokine IL-6, reflecting HSC activation and an inflammatory fibrotic response. Free SCU treatment slightly attenuated the expression of these proteins (e.g., weaker α-SMA and IL-6 bands), but these reductions were not statistically significant. In contrast, SCE markedly decreased the expression of all three markers. Quantification revealed that SCE lowered α-SMA and collagen I protein levels by 47% and 63%, respectively, compared to BDL (*p* < 0.01), indicating substantial inhibition of stellate cell activation and extracellular matrix production. SCE also reduced IL-6 to near-Sham levels, highlighting its potent anti-inflammatory effect. The differences between SCE and free SCU were significant (*p* < 0.05), demonstrating that the nanoemulsion achieved excellent molecular-level therapeutic impact.

Hepatic mRNA levels mirrored the protein results. BDL strongly increased collagen I and α-SMA transcripts (Figure 8D). Free SCU modestly decreased these mRNA levels, with some reductions reaching statistical significance, suggesting a moderate effect on fibrogenic gene expression. SCE, however, produced the strongest suppression: it significantly downregulated collagen I and α-SMA mRNA compared to both BDL and free SCU groups, confirming potent inhibition of fibrogenesis at the transcriptional level. These findings indicate that SCE not only reduces existing fibrotic protein deposition but also actively suppresses ongoing fibrogenic signaling in the liver.

Immunofluorescence staining provided spatial context to these molecular findings (Figure 8E–G). In the BDL model liver sections, α-SMA-positive activated HSCs (green fluorescence) densely lined the fibrotic septa, and collagen I (red fluorescence) was extensively deposited, co-localizing in fibrous strands. Free SCU treatment slightly reduced the number of α-SMA-positive cells and modestly decreased collagen I intensity, but fibrotic streaks remained prominent. In contrast, SCE-treated liver sections had markedly weaker fluorescence for both α-SMA and collagen I. Only a few scattered α-SMA-positive cells were detected, and collagen I staining appeared faint and confined to periportal zones, signifying a markedly reduced fibrosis. Image analysis of the fluorescent areas indicated SCE significantly cut down α-SMA and collagen I positivity compared to BDL (*p* < 0.01), whereas free SCU had a smaller effect. These observations visually confirm that SCE effectively inactivates HSCs and reduces scar matrix in vivo.

Hydroxyproline quantification further supported these results (Figure 8H). BDL significantly elevated hepatic hydroxyproline content, reflecting excessive collagen accumulation. Free SCU slightly reduced hydroxyproline levels by approximately 16%, but the change was not statistically significant, consistent with the modest histological improvements. In contrast, SCE reduced hepatic hydroxyproline content by roughly 71% compared to BDL (*p* < 0.01), approaching the levels observed in Sham mice. These data indicate that SCE substantially curbed collagen accumulation and may have even promoted partial resorption of existing scar tissue, highlighting the formulation’s superior efficacy.

Collectively, these evaluations demonstrate that SCE elicits certain anti-fibrotic effects. Compared to free SCU, SCE more effectively reduces histological fibrosis, lowers fibrotic scar collagen, deactivates HSCs, and improves liver function. The superior therapeutic outcome is likely attributable to enhanced bioavailability and targeted delivery of SCU achieved via the nanoemulsion system. These findings highlight the promise of the SCE formulation as a potent anti-fibrotic intervention.

### 2.7. SCE Safety and Biocompatibility

Beyond efficacy, we assessed the safety of the SCE formulation, which is critical for its potential clinical application.

In vitro cytotoxicity tests using the CCK-8 assay indicated that SCE is non-toxic to both target cells (LX-2 cells) and off-target cells (LO2 hepatocytes) at relevant concentrations (Figure 9A,B). LO2 cells incubated with either free SCU or SCE (at 5–30 µM SCU equivalents) maintained high viability (consistently above 85%). At the highest concentration (30 µM), SCE-treated LO2 cells showed slightly higher viability than those treated with free SCU. LX-2 cells exhibited a similar trend: free SCU did not reduce viability significantly, and SCE-treated cells maintained or slightly exceeded 100% viability relative to control. These data confirm that the nanoformulation itself (composed of phospholipid, MCT oil, Cremophor, and chitosan oligosaccharide) is biocompatible with liver cells and that SCU at therapeutic doses is safe for normal hepatocytes.

For in vivo safety assessment, we administered daily doses of free SCU or SCE (10 mg/kg SCU) to healthy ICR mice (non-BDL) for 14 days, mimicking the treatment regimen. Serum analyses (Figure 9C–F) showed no significant differences among saline control, free SCU, and SCE groups in liver and kidney function markers. ALT and AST remained in the normal low range for all groups, indicating that neither SCU nor SCE caused hepatocellular injury in normal mice. Similarly, triglyceride (TG) levels were not elevated; in fact, they were similar across groups, suggesting the lipid-based nanoemulsion did not disrupt lipid metabolism. Renal function markers, including blood urea nitrogen (UREA) and creatinine (Cre) levels, were also comparable among the groups, implying that renal function was unaffected and there was no nephrotoxicity. These results suggest that SCE does not produce adverse effects on critical organ functions at the given dose.

Histopathological examination supported the safety profile of SCE (Figure 9G). H&E staining of major organs—including the heart, liver, spleen, lungs, and kidneys—revealed no pathological changes in SCE-treated mice. Specifically, liver sections displayed normal lobular architecture with no signs of inflammation or degeneration; kidney glomeruli and tubules were intact; cardiac myocytes, pulmonary alveoli, and splenic white pulp/red pulp were all unremarkable and similar to controls. We observed no signs of organ toxicity, such as cellular infiltration, tissue damage, or lipid accumulation, after SCE administration. Notably, although some nanoemulsion components (e.g., Cremophor EL) can occasionally provoke systemic reactions, we detected no such adverse effects at the administered dose.

In summary, both SCU and its SCE demonstrated excellent safety and tolerability in vitro and in vivo. The nanoemulsion did not introduce any detectable toxicity. These findings indicate a favorable therapeutic index for SCE, meaning we can achieve efficacious concentrations in the target organ (liver) without causing harm to normal cells or other organs. This is a key requirement for advancing such a therapy towards clinical consideration.

## 3. Discussion

Liver fibrosis is characterized by excessive deposition of extracellular matrix components, particularly collagen, leading to distortion of liver architecture and function. If unresolved, fibrosis progresses to cirrhosis and eventually hepatocellular carcinoma [38]. Although researchers have extensively elucidated the mechanisms of fibrogenesis, clinically approved anti-fibrotic therapies remain scarce. Current treatment strategies primarily target underlying etiologies (e.g., antiviral therapy for viral hepatitis) and provide supportive care, but they show limited efficacy in reversing established fibrosis.

SCU has emerged as a potential multi-target agent due to its broad pharmacological activities. Prior studies have documented its anti-fibrotic properties in non-hepatic tissues. In this context, multi-target approaches that simultaneously address different aspects of fibrogenesis are highly desirable. Our study focused on scutellarin (SCU) because of its well-documented pharmacological profile and its critical limitation of poor solubility and bioavailability, which made it an ideal candidate to demonstrate the utility of our nanoemulsion platform. Other flavonoids such as silybin and quercetin are also promising anti-fibrotic candidates, and the nanoemulsion strategy described here could, in principle, be extended to these compounds in future work.

Scutellarin (SCU) has emerged as a promising candidate due to its multitarget pharmacological profile. Previous studies have shown that SCU reduces fibrosis in non-hepatic organs [16,17] and protects the liver [9,10]. Other researchers demonstrated that SCU reshapes gut microbiota in liver disease models [5], matters because gut-derived factors drive liver inflammation. In this study, we showed that SCU treatment altered both intestinal and hepatic microbiota under fibrotic conditions to some extent. SCU reduced the abundance of several potentially pathogenic taxa (such as *Actinobacteriota* and *Desulfobacterota* in the gut), which BDL had elevated. Elevated *Actinobacteriota* abundance worsens liver pathology in steatosis and cancer models [19,20], whereas elevated *Desulfobacterota* abundance is linked to hepatic inflammation [21]. Although the nanoemulsion primarily facilitated efficient hepatic targeting, its oral route of administration also allowed for transient interaction with the gut microbiota. This was associated with partial normalization of dysbiotic taxa, which—while not the principal mechanism of action—may have contributed additively to the observed therapeutic benefits. By attenuating these dysbiotic shifts, SCU likely blocked the translocation of pro-fibrogenic microbial products such as endotoxins from the gut to the liver. SCU also restored normal levels of *Erysipelotrichales* and *Oscillibacter* in the hepatic microbiota, which indicates that SCU directly or indirectly regulates bacteria that colonize or translocate to the liver during fibrogenesis. The liver microbiome is a relatively new research frontier. We observed that SCU shifted microbial abundance, but we have not yet determined how these changes drive anti-fibrotic outcomes. The liver microbiota results were obtained from low-biomass samples and should therefore be regarded as highly exploratory and inherently prone to contamination—a well-recognized limitation in this field. Although we implemented stringent precautions, including processing all samples under a biosafety cabinet with sterile, DNA-free reagents, the absence of dedicated negative controls and formal bioinformatic decontamination necessitates cautious interpretation. Accordingly, we do not present these findings as definitive evidence of a resident liver microbiota, but rather as preliminary, hypothesis-generating observations that are consistent with the emerging concept of a gut–liver axis in fibrosis. We classify our hepatic microbiota findings as exploratory. Future studies using germ-free or microbiota-depleted models containing adequate mice must establish whether microbiome modulation directly improves fibrosis. Despite this limitation, our results strengthen the concept of a gut–liver axis in fibrosis and show that SCU restores microbial balance to improve this axis. Furthermore, we acknowledge that microbiota abundance data may not follow a normal distribution, which limits the robustness of parametric comparisons.

SCU shows poor water solubility and low oral bioavailability, which likely hinders its therapeutic effectiveness in vivo [12,18]. In this study, we overcame this barrier by developing the scutellarin–phospholipid complex (SPC). Characterization techniques, including FTIR, DSC, and XRD, confirmed that SCU was present in an amorphous state within the lipid matrix, likely at least partially molecularly dispersed. Amorphization is known to enhance the dissolution of hydrophobic drugs [39]. By incorporating SPC into a nanoemulsion (SCE), we formulated SCU at therapeutically relevant concentrations with greater solubility and absorption. Nanoemulsions commonly increase the oral bioavailability of lipophilic compounds by promoting lymphatic transport and preventing precipitation or metabolism [40]. Our findings support this mechanism: the SCE nanoemulsion likely promoted intestinal absorption via lymphatic transport pathways, thereby enhancing hepatic delivery of SCU, as corroborated by fluorescence imaging.

SCE significantly enhanced the pharmacokinetic and tissue distribution profiles of SCU. In vivo imaging results showed that SCE achieved higher and more sustained liver concentrations than free SCU. This liver-targeting effect benefits anti-fibrotic therapy by concentrating the drug at the site of action and reducing systemic exposure, thereby lowering the risk of off-target effects. SCE also prolonged hepatic retention, maintaining significant levels even 24 h after treatment. This sustained presence may extend drug action and permit less frequent dosing in clinical applications.

At the cellular level, the nanoemulsion also proved advantageous. Our results showed that SCE is readily taken up by liver cells (stellate cells, hepatocytes, and possibly liver macrophages) once it reaches the liver. The mechanism involved caveolae-mediated endocytosis, a pathway that nanoparticles often exploit to enter cells efficiently. Through this process, the nanoemulsion allowed SCU to cross the cellular barriers of fibrotic tissue more effectively. In fibrotic livers, the dense extracellular matrix typically impedes drug diffusion. However, nanoemulsions of suitable size can exploit disrupted sinusoidal endothelium and increased vascular permeability to access activated HSCs, which overexpress endocytic receptors. This mechanism likely explains why SCE suppressed HSC activation more effectively than free SCU in vivo—greater intracellular delivery enabled the drug to exert its pharmacological effect.

Therapeutically, SCE treatment produced certain outcomes in the BDL-induced fibrosis model. SCE improved the progression of fibrosis, as indicated by the lower collagen content and improved histology compared to untreated fibrotic mice. In contrast, free SCU exhibited only minimal effects, underscoring the critical role of the nanoemulsion delivery system in unlocking the therapeutic potential of SCU. The limited efficacy of free SCU at a dose of 10 mg/kg is consistent with previous reports that its poor bioavailability hinders in vivo activity [12,41,42]. Using SCE, we effectively increased the bioavailability and hepatic concentration of SCU, thereby achieving the desired anti-fibrotic action.

Mechanistically, SCU exerts anti-fibrotic effects through several interconnected pathways. Known for its antioxidant and anti-inflammatory properties [10], SCU disrupts TGF-β/SMAD signaling, the central fibrogenic pathway, and downregulates key fibrotic markers such as α-SMA and collagen I. SCU also modulates inflammatory cascades, likely through NF-κB signaling, as evidenced by the pronounced reduction in hepatic IL-6 expression in SCE-treated mice. The downregulation is particularly significant, since IL-6 not only signals inflammation but also drives fibrosis and carcinogenesis in chronic liver disease. By suppressing IL-6, SCU helps dampen the inflammatory milieu that fuels fibrogenesis. We also found that SCU reduced MMP2 expression in HSCs in vitro. This modulation of MMP2 suggests that SCU influences matrix remodeling dynamics, promoting a more balanced environment where collagen deposition and degradation can proceed toward the resolution of fibrosis.

Our study also highlights the favorable safety of the SCE system. We were careful to demonstrate that the formulation components (lipid, surfactant, and COS polymer) did not introduce toxicity. COS is generally regarded as biocompatible and has been reported to confer additional biological benefits, such as promoting intestinal health and enhancing mucosal permeability. The surfactant (Cremophor EL), while sometimes causing hypersensitivity at high doses in intravenous formulations, is in a relatively low dose orally and was well-tolerated. Furthermore, the negative surface charge and nanoscale size of SCE contributed to its stability and low immune recognition, preventing unwanted immune responses.

From a translational perspective, SCE shows promise as a therapy for liver fibrosis. Nevertheless, several considerations and future directions must be addressed. First, although our results in the BDL model were compelling, liver fibrosis is a heterogeneous condition with diverse etiologies. Future work should evaluate the efficacy of SCE in additional models, such as those induced by carbon tetrachloride or NASH, to confirm its broader applicability. Second, the observed microbiome-modulating effects of SCU raise the possibility of combining SCU with specific probiotics or prebiotics to enhance therapeutic outcomes via synergistic modulation of the gut–liver axis. Third, while nanoemulsions are relatively straightforward to scale up, we must confirm long-term stability beyond seven days. Transforming the nanoemulsion into a solid dosage form, such as a freeze-dried powder for reconstitution, could improve practicality for clinical use. Fourthly, another limitation of this study is the use of a single dose of SCE. While this dose was effective and informed by previous studies [41,42], future work will include a comprehensive dose–response evaluation to determine the optimal therapeutic window and maximize the potential of this formulation. Finally, another limitation of this study is the absence of a blank nanoemulsion control in the in vitro experiments. Although our primary objective was to evaluate the therapeutic potential of SCU when formulated into a nanoemulsion, we acknowledge that vehicle-only controls would have further strengthened the interpretation of these findings. Without this control, we cannot fully exclude the possibility that some of the observed effects may be partially attributable to the components of the vehicle rather than SCU itself. Future studies will incorporate such vehicle controls to rigorously confirm the specificity of the nanoemulsion-mediated effects.

In addition, we acknowledge that effect sizes and confidence intervals were not provided, which limits the interpretability of some results.

The strategy of targeting both fibrotic processes and the microbiome is still relatively novel. Our work provides proof-of-concept that a single agent, when properly formulated, may simultaneously modulate HSC activity and the gut–liver axis. SCU exerts a dual mechanism of action by directly suppressing fibrotic processes and simultaneously ancillary modulating the microbiota, thereby exemplifying a polypharmacological strategy [7]. In complex diseases such as liver fibrosis, where inflammation, cell activation, and gut–liver signaling interact through tightly linked pathways, multi-pronged agents like SCU achieve greater therapeutic effectiveness.

In conclusion, this work demonstrates that formulating SCU into a nanoemulsion markedly enhances its therapeutic efficacy against liver fibrosis. SCE improves pharmacokinetics and liver targeting, which increases anti-fibrotic effects by reducing collagen deposition and HSC activation. Additionally, SCE preserves the ability of SCU to modulate the gut–liver axis through microbiota changes. These results highlight the critical role of drug delivery systems in unlocking the clinical potential of poorly soluble natural products like SCU. The nanoemulsion approach described herein may apply to other phytochemicals or therapeutic agents facing similar solubility and bioavailability challenges. Overall, our findings contribute to the development of an effective, multi-target therapy for liver fibrosis and highlight the innovative angle of targeting the microbiome as part of the therapeutic mechanism.

## 4. Materials and Methods

### 4.1. Materials

Scutellarin (SCU, >98% purity, CAS No. 27740-01-8) was purchased from J&K Scientific (Wuhan, China). Soybean phospholipid (lecithin) served as the excipient for phospholipid complex preparation. Caprylic/capric triglycerides (medium-chain triglycerides) and Cremophor EL (polyoxyethylated castor oil) were used as the oil phase and surfactant, respectively, in the nanoemulsion. Chitosan oligosaccharide (COS) was used as a stabilizer. Recombinant human TGF-β1 (transforming growth factor beta 1) was purchased from R&D Systems (Minneapolis, MN, USA) for HSC activation in vitro. The human HSC line LX-2 was purchased from Shanghai Mingjin Biotechnology Co., Ltd. (Shanghai, China) (RRID:CVCL_5792) on 16 May 2023. The human normal liver cell line LO2 was kindly provided by Professor He (Institute of Medicinal Biotechnology, Chinese Academy of Medical Sciences & Peking Union Medical College). Antibodies against α-SMA (α-smooth muscle actin, Cat# 14395-1-AP), MMP2 (matrix metalloproteinase-2, Cat# 66366-1-Ig), and collagen type I (COL1A1, Cat# 14695-1-AP) were purchased from Proteintech (Wuhan, China). GAPDH antibody (Cat# ABL-1021) was purchased from Abbkine (Wuhan, China). All other reagents and chemicals were of analytical reagent grade and were used as received without further purification.

### 4.2. Preparation of Scutellarin–Phospholipid Complex (SPC)

A scutellarin–phospholipid complex (SPC) was prepared to enhance the lipophilicity of SCU, following a previously reported method with slight modifications [39]. Briefly, SCU and soybean phospholipid (mass ratio 1:5) were co-dissolved in absolute ethanol to obtain a solution containing 0.5 mg/mL of SCU. The solvent was subsequently removed by rotary evaporation under reduced pressure at approximately 40 °C. During evaporation, the mixture was sonicated to facilitate molecular interaction between SCU and the phospholipid. The resulting solid residue was dried to constant weight to yield the SPC. For comparison, a physical mixture of SCU and phospholipid was prepared by manually blending the two components at the same 1:5 mass ratio using a mortar and pestle, without the use of any solvent.

### 4.3. Preparation of SCU-Loaded Nanoemulsion (SCE)

The SCU-loaded nanoemulsion (SCE) was prepared using the previously prepared SPC. First, a primary coarse emulsion was prepared using the phase inversion method. Caprylic/capric triglyceride (1 mL) was mixed with an equal mass of Cremophor EL to form the oil-surfactant phase. Subsequently, 48 mg of SPC (containing SCU) was added to the oil-surfactant phase and dissolved with gentle heating and sonication until a clear solution was obtained. Separately, chitosan oligosaccharide (15 mg) was dissolved in 3 mL of deionized water to prepare the aqueous phase. The oil phase was added dropwise into the aqueous phase under magnetic stirring at approximately 1500 rpm at room temperature, resulting in a crude oil-in-water emulsion. This primary emulsion was then subjected to high-energy ultrasonication to reduce droplet size, using an ultrasonic cell disruptor (Xinzhi, Ningbo, China) (200 Hz) in an ice-water bath with pulse cycles of 10 s on and 5 s off for a total of 15 min. The resulting nanoemulsion was equilibrated to room temperature and stored at 4 °C until use.

### 4.4. Characterization of SPC and SCE

To confirm SPC formation, differential scanning calorimetry (DSC), Fourier-transform infrared spectroscopy (FTIR), and X-ray powder diffraction (XRD) analyses were conducted on SCU, phospholipid, their physical mixture, and the SPC product. DSC thermograms were acquired using a DSC analyzer (Mettler-Toledo, Switzerland) by heating samples from 30 °C to 400 °C at a programmed rate to detect alterations in melting or crystallization behavior indicative of complex formation. FTIR spectra were recorded using a Nicolet 5700 FTIR spectrometer (Thermo, Waltham, MA, USA) over the range 4000–400 cm^−1^ to identify chemical interactions, such as hydrogen bonding, between SCU and the phospholipid. XRD patterns were recorded using a Bruker D8 Advance diffractometer (Billerica, MA, USA) (Cu Kα radiation, λ = 1.5406 Å, 40 kV, 40 mA) to assess the crystallinity of SCU in each sample; the disappearance of SCU’s characteristic crystalline peaks in the SPC would suggest an amorphous or molecularly dispersed state.

SCE was characterized in terms of particle size, size distribution, zeta potential, morphology, and drug content. The hydrodynamic particle size (mean diameter) and polydispersity index (PDI) were measured by dynamic light scattering (DLS) using a Zetasizer Nano ZS (Malvern Instruments, Malvern, UK) at 25 °C. Zeta potential, indicating the surface charge of nanoemulsion droplets, was determined by electrophoretic light scattering with the same instrument. The morphology of SCE was observed using transmission electron microscopy (TEM). A drop of diluted SCE was placed on a carbon-coated copper grid, allowed to sit for 1 min, and excess fluid was removed by blotting. The grid was then air-dried and examined under a TEM (JEM-2100, JEOL, Akishima, Japan) operated at an accelerating voltage of 200 kV. The TEM images provided visual confirmation of particle size and morphology. The SCU concentration in the nanoemulsion was determined using a validated high-performance liquid chromatography (HPLC) method (Shimadzu, Kyoto, Japan) with UV detection at 335 nm. Briefly, SCE samples were diluted in methanol and analyzed on a C18 column. The mobile phase consisted of methanol and 0.5% acetic acid (4:6, *v*/*v*) and was delivered at a flow rate of 1 mL/min. The column temperature was maintained at 30 °C. Quantification was achieved using a standard calibration curve.

### 4.5. Storage Stability Study

The short-term physical and chemical stability of SCE was assessed over 7 days at 4 °C. Aliquots were stored in sealed vials and sampled on days 0 (initial), 1, 3, 5, and 7. At each time point, mean particle size, PDI, and zeta potential were measured by DLS as described in Section 4.4. Additionally, the SCU concentration within the formulation was analyzed by HPLC to monitor any potential drug degradation or precipitation. All measurements were performed in triplicate. Stability was considered acceptable if no significant particle growth or aggregation occurred and if the SCU content remained above 90% of the initial drug content over the storage period.

### 4.6. Cell Culture and Cytotoxicity Assay

LX-2 and LO2 cells were cultured in Dulbecco’s Modified Eagle Medium (DMEM, Gibco, Waltham, MA, USA) and Roswell Park Memorial Institute (RPMI) 1640 (Gibco, USA), respectively. Both media were supplemented with 10% fetal bovine serum (FBS) and 1% penicillin–streptomycin. Cells were maintained at 37 °C in a humidified atmosphere with 5% CO_2_. The culture medium was replaced every 2–3 days, and cells were subcultured using trypsin-EDTA upon reaching 80–90% confluence.

The cytotoxicity of SCE was evaluated in vitro using the Cell Counting Kit-8 (CCK-8, Meilunbio, Dalian, China) assay in both LX-2 and LO2 cell lines. Cells were seeded into 96-well plates at a density of 5 × 10^3^ cells per well and allowed to adhere for 12 h. Subsequently, the medium was replaced with 100 µL of fresh medium containing either free SCU or SCE at various concentrations (ranging from 5 µM to 30 µM SCU equivalent). Control wells received only drug-free medium. After 24 h of treatment, 10 µL of CCK-8 reagent was added to each well, followed by an additional 4 h incubation period. Absorbance was measured at 450 nm using a microplate reader (Synergy H1, BioTek, Winooski, VT, USA). Cell viability was expressed as a percentage of the untreated control. Each concentration was tested in quadruplicate, and data are presented as mean ± SD. For all cell-based assays (cytotoxicity, cellular uptake, gene/protein expression, and migration), at least three independent experiments were performed on different days, and each experiment contained multiple technical replicates.

### 4.7. Cellular Uptake Study in LX-2 Cells

The cellular uptake of SCU delivered via SCE versus free drug was examined in LX-2 cells using a fluorescent probe. Nile Red (NR), a hydrophobic fluorescent dye, was used as a surrogate to visualize and quantify uptake. NR-loaded SCE was prepared by adding a small amount of NR to the SPC (at 10% *w*/*w* of SCU) during the nanoemulsion preparation process (Section 4.3). A solution of free NR in PBS (containing a small amount of DMSO to aid solubilization) was used as a control corresponding to free SCU.

For qualitative uptake visualization, LX-2 cells were seeded in glass-bottomed confocal dishes at 2 × 10^5^ cells per dish and cultured for 24 h. Cells were then treated with either free NR (in PBS) or NR-loaded SCE (NR/SCE) at an equivalent NR concentration. After incubation for predetermined periods (5 min, 15 min, 30 min, 1 h, 2 h, and 4 h) at 37 °C, the cells were washed three times with PBS to remove extracellular NR, fixed with 4% paraformaldehyde for 15 min, and stained with DAPI (4′,6-diamidino-2-phenylindole) to label nuclei. Fluorescent images were acquired using a laser scanning confocal microscope (Zeiss LSM 710, Oberkochen, Germany). NR (red) and DAPI (blue) signals were visualized to evaluate intracellular localization and uptake intensity.

For quantitative analysis, cellular uptake was evaluated by flow cytometry. LX-2 cells were seeded in 6-well plates (3 × 10^5^ cells/well) and treated with free NR or NR-loaded SCE as described above. At selected time points (up to 4 h), cells were washed with ice-cold PBS, detached with trypsin, and resuspended in PBS. Intracellular fluorescence was measured using a flow cytometer (BD FACSCalibur, Franklin Lakes, NJ, USA) in the FL2 channel. At least 10,000 events were recorded per sample. The mean fluorescence intensity (MFI) was calculated to compare the cellular uptake efficiency of NR/SCE versus free NR.

To elucidate the cellular internalization pathways of SCE, LX-2 cells were pre-treated for 15 min with one of several pharmacological endocytosis inhibitors before exposure to NR-loaded SCE (NR/SCE). The inhibitors included: chlorpromazine (10 µg/mL, inhibitor of clathrin-mediated endocytosis), nystatin (25 µg/mL, inhibits caveolae-mediated endocytosis by cholesterol sequestration), methyl-β-cyclodextrin (MβCD, 5 mM, depletes membrane cholesterol and disrupts caveolae), genistein (100 µM, tyrosine kinase inhibitor that also disrupts caveolae-mediated endocytosis), imipramine (10 µg/mL, reported to inhibit caveolae pathway), and 5-(N-ethyl-N-isopropyl) amiloride (EIPA, 50 µM, inhibitor of macropinocytosis). Following inhibitor pretreatment, cells were washed with PBS and incubated with NR/SCE for 2 h at a concentration previously determined to produce intense fluorescence. Cells were then processed for flow cytometry as described in the cellular uptake section. A significant reduction in MFI in the presence of a specific inhibitor, relative to the untreated control, was interpreted as indicative of that pathway’s involvement in SCE internalization.

### 4.8. In Vitro Anti-Fibrotic Activity in LX-2 Cells

LX-2 cells were employed as an in vitro model to assess HSC activation and evaluate the anti-fibrotic potential of SCU formulations. Cells were seeded in 6-well plates at a density of approximately 1 × 10^5^ cells per well and cultured to 80–90% confluence. To induce a fibrogenic phenotype, the cells were serum-starved in DMEM containing 2% FBS for 24 h, followed by stimulation with TGF-β1 (2 ng/mL) for an additional 24 h. This treatment activates LX-2 cells, resulting in upregulation of fibrotic markers. After activation, the medium was replaced, and SCE was added. The cells were then incubated for 12 h. Subsequently, the cells were harvested for analysis of fibrogenic gene and protein expression by quantitative real-time PCR (qPCR) and Western blot.

Total RNA was isolated using the RaPure Total RNA Kit (Magen, Guangzhou, China) according to the manufacturer’s protocol. RNA concentration and purity were verified by spectrophotometry. cDNA was synthesized and amplified using a one-step RT-qPCR SYBR Green kit (Vazyme, Nanjing, China) on a 7500 Fast Real-Time PCR System (Applied Biosystems, Waltham, MA, USA). The primer sequences for target genes (collagen I [COL1A1], α-SMA [ACTA2], and β-actin [ACTB] as a housekeeping gene) were as follows: COL1A1: forward 5′-TGACCTTCCTGCGCCTAATG-3′, reverse 5′-GCTACGCTGTTCTTGCAGTG-3′; ACTA2 (α-SMA): forward 5′-CTCTGTCTGGATCGGTGGC-3′, reverse 5′-TTCGTCGTATTCCTGTTTGCT-3′; ACTB: forward 5′-CCTGGACTTCGAGCAAGAGATGG-3′, reverse 5′-GTGGTTTCGCTCGGCACATT-3′. The thermal cycling conditions were set according to kit protocols. The relative gene expression was calculated using the 2^(−ΔΔCt)^ method, with normalization to ACTB and comparison to the untreated control group.

For Western blot analysis, we lysed LX-2 cells using radioimmunoprecipitation assay (RIPA) buffer supplemented with protease and phosphatase inhibitors to extract total protein. We measured protein concentrations and loaded equal amounts (20 µg per sample) onto SDS-PAGE gels, then transferred the separated proteins onto polyvinylidene difluoride (PVDF) membranes. We blocked the membranes with 5% non-fat milk for 1 h, followed by overnight incubation at 4 °C with primary antibodies against key fibrogenic proteins: collagen I (1:2000 dilution) and MMP2 (1:2000). GAPDH (1:5000) was used as an internal loading control. After washing, we incubated the membranes with horseradish peroxidase-conjugated secondary antibodies (1:10,000) for 1 h at room temperature. We visualized protein bands using enhanced chemiluminescence (ECL) substrate and captured images with a ChemiDoc imaging system (Bio-Rad, Hercules, CA, USA). We quantified band intensities using ImageJ software (1.54) and normalized target protein levels to GAPDH.

### 4.9. Cell Migration Assay

We evaluated the effect of SCU on HSC migration, a characteristic of activated HSCs that contributes to fibrotic tissue remodeling, using a wound healing (scratch) assay. LX-2 cells were seeded in 6-well plates and cultured until they formed a nearly confluent monolayer. We created a linear scratch approximately 1 mm wide through the cell monolayer using a sterile pipette tip. After gently washing the wells with PBS to remove detached cells and debris, we treated the remaining cells in serum-reduced medium containing TGF-β1 and SCU. We captured images of the wound area at 0 h (immediately after scratching), 12 h, and 24 h post-treatment using an inverted phase-contrast microscope. We quantified cell migration by measuring the remaining wound width or calculating the wound area at each time point relative to the initial wound area at 0 h.

### 4.10. In Vivo Tissue Distribution and Liver Targeting

We assessed the liver-targeting efficiency of SCE using a near-infrared fluorescent probe, DIR (1,1′-dioctadecyl-3,3,3′,3′-tetramethylindotricarbocyanine iodide), which is lipophilic and suitable for tracking nanoemulsion distribution via fluorescence imaging. We prepared DIR-loaded SCE similarly to NR/SCE by dissolving DIR in the oil phase before emulsification. As a control, we prepared free DIR by dissolving the dye in a Cremophor/ethanol mixture and subsequently diluting it with saline to mimic the free drug formulation.

Male ICR mice (6–8 weeks old, approximately 25 g) were obtained from Beijing Vital River Labs (Beijing, China) and acclimatized for one week under standard laboratory conditions. All animal experiments were approved by the Institutional Animal Care and Use Committee (IACUC) of the Institute of Medicinal Biotechnology, CAMS & PUMC (Approval No. IMB-20231109D102), and conducted per national ethical guidelines. The mice were fasted for 12 h before the experiment and randomly divided into two treatment groups (n = 3 per group per time point): free DIR and DIR-loaded SCE groups. Each mouse received a single oral gavage of DIR (0.5 mg/kg). At 1, 3, 6, 12, and 24 h post-administration, we anesthetized and euthanized three mice from each group and excised their major organs (heart, liver, spleen, lung, and kidneys). We immediately performed ex vivo fluorescence imaging using the IVIS system (PerkinElmer, Waltham, MA, USA) with DIR-appropriate filter settings. We drew regions of interest (ROI) over each organ to quantify fluorescence intensity (radiant efficiency), with particular focus on the liver to assess targeting efficiency. The mean fluorescence intensity of the liver at each time point was calculated and compared between the two groups to evaluate the extent and duration of hepatic accumulation under the appropriate excitation/emission filter settings for DIR.

### 4.11. Bile Duct Ligation-Induced Liver Fibrosis Model and Treatment Protocol

We employed a bile duct ligation (BDL) mouse model to induce liver fibrosis and evaluate the therapeutic effects of SCU formulations. Male ICR mice (6–8 weeks old) were randomly divided into four groups (n = 5–6 per group): sham (sham operation + vehicle treatment), BDL model (BDL + vehicle), free SCU (BDL + free SCU treatment), and SCE (BDL + SCU nanoemulsion treatment). The BDL procedure was performed under anesthesia (isoflurane, gas anesthesia) by double ligation and transection of the common bile duct using sterile technique. In the sham-operated group, we exposed the bile duct but left it intact. Following surgery, we administered buprenorphine for postoperative analgesia and monitored all animals closely until full recovery. Each experimental group included 5 animals (biological replicates). For each endpoint, the exact number of samples analyzed is indicated in the figure legends. All data are presented as mean ± SD, and statistical analyses were performed using the full dataset (n = 5), unless otherwise specified.

We began treatment on postoperative day 2. Free SCU was freshly prepared by dissolving scutellarin in saline containing a minimal amount of NaOH to enhance solubility. This solution was then administered at a dose of 10 mg/kg. We formulated SCE to deliver an equivalent SCU dose. The Sham and BDL control groups received an equivalent volume of normal saline. We administered all treatments once daily by oral gavage for 14 consecutive days, recorded body weights regularly, and noted any signs of distress.

At the end of the treatment period, we anesthetized the mice and collected blood samples via the orbital sinus. Serum was separated by centrifugation at 3500 rpm for 10 min at 4 °C and stored at −80 °C for subsequent biochemical analyses. We then euthanized the mice and harvested liver tissues. Portions of the liver were fixed in 10% neutral-buffered formalin for histological and immunofluorescence analysis, whereas other portions were snap-frozen in liquid nitrogen and stored at −80 °C for biochemical assays, including Western blotting and hydroxyproline quantification. Additionally, we aseptically collected 50 mg of fresh liver tissue and fecal pellets from the colon and immediately frozen for microbiome analysis via 16S rRNA gene sequencing.

We performed quantitative real-time PCR (qPCR) and Western blot analyses to assess the expression of fibrogenic genes and proteins in liver tissues, following the procedures described in Section 4.8. Notably, the primer sequences used for liver tissue qPCR differed from those used for cellular analysis. The primer sequences were as follows: COL1A1: forward 5′-CATGTTCAGCTTTGTGGACCT-3′, reverse 5′-GCAGCTGACTTCAGGGATGT-3; ACTA2: 5′-TTCCTTCGTGACTACTGCCG-3′, reverse 5′-TATAGGTGGTTTCGTGGATGCC-3′; ACTB: forward 5′-CGTTCAATACCCCAGCCATG-3′, reverse 5′-GACCCCGTCACCAGAGTCC-3′.

### 4.12. Serum Biochemistry Analysis

To evaluate liver injury and systemic responses to treatment, we quantified serum levels of key liver function markers. We measured alanine aminotransferase (ALT), aspartate aminotransferase (AST), and alkaline phosphatase (ALP) activities using an automated biochemical analyzer or colorimetric assay kits (Nanjing Jiancheng Bioengineering, Nanjing, China) following the manufacturer’s protocols. We also assessed serum total bile acids (TBA) as an indicator of BDL-induced cholestasis. We performed all assays in duplicate, and results were reported as mean ± SD for each group.

### 4.13. Histological Analysis

For histopathological examination, liver and other major organs—including the heart, spleen, lungs, and kidneys—were fixed in formalin, embedded in paraffin, and sectioned at a thickness of 4 µm. We stained liver sections with hematoxylin and eosin (H&E) to evaluate general liver architecture and injury. To visualize collagen deposition and fibrosis, we performed Masson’s trichrome and Sirius Red staining. Masson’s stain renders collagen fibers blue, whereas Sirius Red binds specifically to collagen, appearing red under light microscopy and exhibiting birefringence under polarized light.

To evaluate potential off-target toxicity, we stained heart, lung, spleen, and kidney sections from mice treated with saline, free SCU, or SCE (without BDL surgery) with H&E. A pathologist blinded to the treatment groups examined all tissues for signs of inflammation, necrosis, or other histological abnormalities.

### 4.14. Hydroxyproline Assay

We quantified hepatic hydroxyproline content as a surrogate marker for collagen accumulation and liver fibrosis. For each mouse, we assayed ~50 mg of liver tissue using a commercial hydroxyproline assay kit (Nanjing Jiancheng Bioengineering Institute, China), following the manufacturer’s instructions. Briefly, liver samples were hydrolyzed in concentrated hydrochloric acid at 110 °C for several hours to release free hydroxyproline from collagen. After hydrolysis, the samples were neutralized and treated with, followed by reaction with Ehrlich’s reagent (p-dimethylaminobenzaldehyde) to generate a chromogenic complex. The absorbance of the resulting solution was then measured at 550 nm and compared to a standard hydroxyproline calibration curve. Hydroxyproline content was expressed as micrograms per gram of liver tissue. All measurements were performed in duplicate.

### 4.15. 16S rRNA Gene Sequencing for Microbiota Analysis

We assessed the impact of SCU on gut and liver microbiota composition by performing 16S rRNA gene sequencing on fecal and liver samples from the Sham, BDL, and SCU-treated groups [43,44]. Genomic DNA was extracted from 100 mg of fecal material using the QIAamp Fast DNA Stool Mini Kit (Qiagen, Hilden, Germany) and from approximately 25 mg of liver tissue using the DNeasy Blood & Tissue Kit (Qiagen) per the respective manufacturer’s protocols. We amplified the hypervariable V3–V4 regions of the bacterial 16S rRNA gene by PCR using universal primers (338F/806R) with Illumina adapter overhangs. After purification with the AxyPrep DNA Gel Extraction Kit (Axygen, Corning, NY, USA), we quantified the PCR products. Equimolar amounts of each sample’s PCR product were pooled to construct sequencing libraries using the TruSeq DNA Sample Prep Kit (Illumina, San Diego, CA, USA). High-throughput sequencing was performed on an Illumina HiSeq 2500 platform, generating paired-end reads of 2 × 250 bp.

We processed raw sequencing reads using the QIIME pipeline. After quality filtering and merging, we removed chimeric sequences to obtain high-quality reads. We clustered the reads into operational taxonomic units (OTUs) at 97% sequence similarity and assigned taxonomic identities to representative sequences using the Greengenes or Silva reference database with the RDP classifier algorithm. To standardize sequencing depth across samples, we normalized the OTU abundance table before downstream analyses. We assessed within-sample microbial diversity using alpha diversity metrics, including Chao1 richness and Shannon diversity indices. To evaluate between-group variation, we analyzed beta diversity by principal coordinates analysis (PCoA) and partial least squares discriminant analysis (PLS-DA) based on Bray–Curtis distances. Finally, we identified differentially abundant taxa at the phylum, family, and genus levels and correlated them with disease status or treatment.

All liver samples were collected under aseptic conditions. DNA extraction and PCR reagents were handled in a sterile environment to minimize contamination. Although no dedicated blank extraction controls were included, all experimental groups were processed in parallel under identical conditions. Data interpretation was based on relative differences between treatment and control groups rather than absolute abundance. Notably, due to the relatively low bacterial biomass in liver tissue, the liver microbiome analysis required careful contamination control and validation, and the results from liver 16S sequencing should be considered exploratory.

### 4.16. Immunofluorescence Staining of Liver Sections

Immunofluorescence was performed to visualize key fibrosis markers (collagen I and α-SMA) in liver tissue sections. Paraffin-embedded liver sections (4 µm) were deparaffinized, rehydrated, and subjected to antigen retrieval (heating in citrate buffer, pH 6.0). After blocking with 5% bovine serum albumin and 0.3% Triton X-100 in PBS for 1 h, sections were incubated overnight at 4 °C with primary antibodies against collagen I (rabbit polyclonal, 1:200 dilution) and α-SMA (mouse monoclonal, 1:200). The next day, sections were washed and incubated for 1 h at room temperature in the dark with species-specific secondary antibodies (Alexa Fluor 594–conjugated goat anti-rabbit for collagen I, Alexa Fluor 488–conjugated goat anti-mouse for α-SMA). Nuclei were counterstained with DAPI, and Slides were mounted with antifade medium before fluorescence microscopy.

### 4.17. Statistical Analysis

For normally distributed data, results are expressed as mean ± SD and analyzed using one-way ANOVA with Tukey’s post hoc test. For data not following a normal distribution (e.g., microbiota abundances), values are presented as median with interquartile range (IQR), and non-parametric tests were applied as appropriate. *p* < 0.05 (corrected with q value if necessary) was considered statistically significant.

## 5. Conclusions

In summary, we developed SCE to overcome the bioavailability limitations of SCU, achieve targeted hepatic delivery, and produce pronounced anti-fibrotic effects. This nanocarrier system enabled SCU to directly inhibit HSC activation and fibrogenic pathways. In vitro, SCE markedly improved SCU uptake by HSCs and suppressed the expression of fibrotic markers, including collagen I and MMP2. In vivo, SCE treatment in BDL-induced fibrotic mice improved histological and biochemical outcomes, reducing collagen deposition, lowering hydroxyproline content, and restoring liver function. The oral administration of SCU also allowed transient interaction with gut microbiota and liver microbiota, partially normalizing dysbiotic taxa. While this effect was not the main mechanism, it likely contributed additively to therapeutic outcomes. Thus, SCE exerts dual-site activity by modulating both hepatic fibrogenesis and gut microbiota imbalance. Collectively, our findings indicate that SCE is a safe and effective nanomedicine with dual activity: targeting both hepatic fibrogenesis and microbiota imbalance. This strategy exemplifies the potential of oral nanoformulations for multi-target treatment of chronic liver diseases, particularly liver fibrosis.

## Figures and Tables

**Figure 1 ijms-26-09746-f001:**
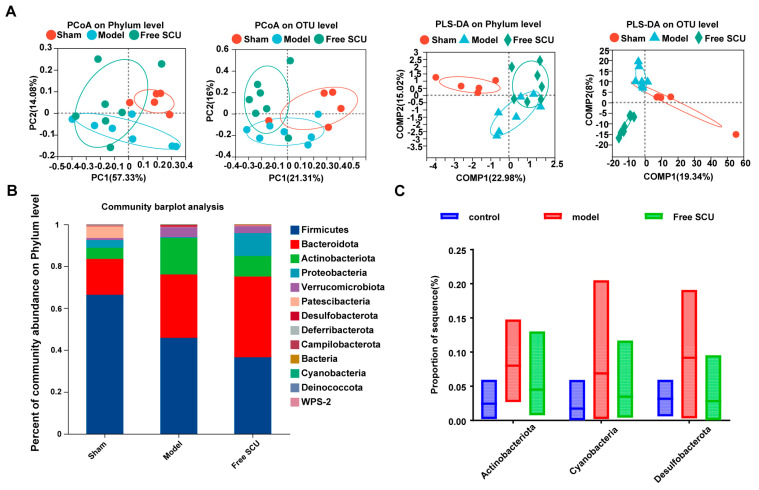
Scutellarin nanoemulsion (SCE) ameliorated BDL-induced gut microbiota dysbiosis. (**A**) Principal Coordinates Analysis (PCoA) plot based on Bray–Curtis distances at the OTU level. (**B**,**C**) Relative abundance of major bacterial phyla in the gut.

**Figure 2 ijms-26-09746-f002:**
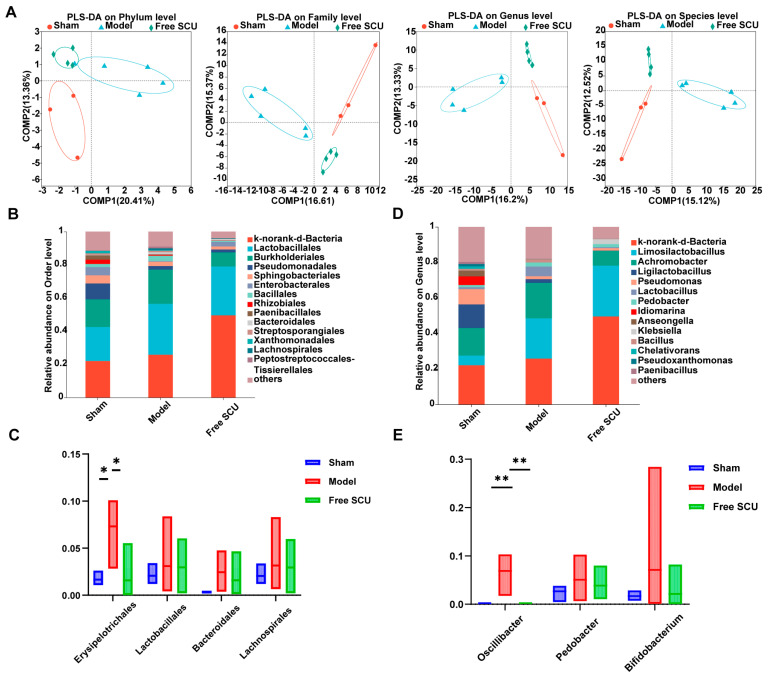
SCU treatment influenced the diversity and composition of the liver microbiome in BDL mice. (**A**) Partial Least Squares Discriminant Analysis (PLS-DA) of liver microbiota profiles at multiple taxonomic levels. (**B**,**C**) Relative abundance of selected bacterial orders in liver tissue. (**D**,**E**) Relative abundance of selected genera in liver microbiota. Liver microbiota data should be interpreted as exploratory due to low biomass. ** *p* < 0.01, * *p* < 0.05.

**Figure 3 ijms-26-09746-f003:**
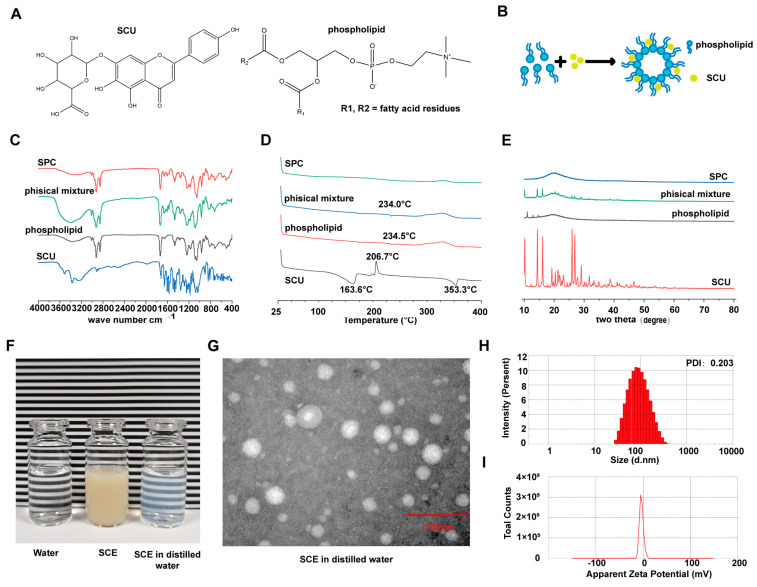
Preparation and physicochemical characterization of the SCU-phospholipid complex (SPC) and the SCU nanoemulsion (SCE). (**A**) Chemical structures of scutellarin (SCU, left) and the phospholipid (soy lecithin, primarily phosphatidylcholine, right) used to form the SPC. (**B**) Schematic illustration of the SPC formation process. (**C**) FTIR spectra of SCU, phospholipid, their physical mixture, and SPC. (**D**) Differential Scanning Calorimetry thermograms of SCU, phospholipid, physical mixture, and SPC. (**E**) X-ray diffraction patterns of the same four samples. (**F**) Photograph of vials containing distilled water, undiluted SCE, and SCE after dilution in water. (**G**) Transmission electron microscopy (TEM) image of SCE. (**H**,**I**) Particle size distribution and zeta potential of SCE measured by DLS.

**Figure 4 ijms-26-09746-f004:**
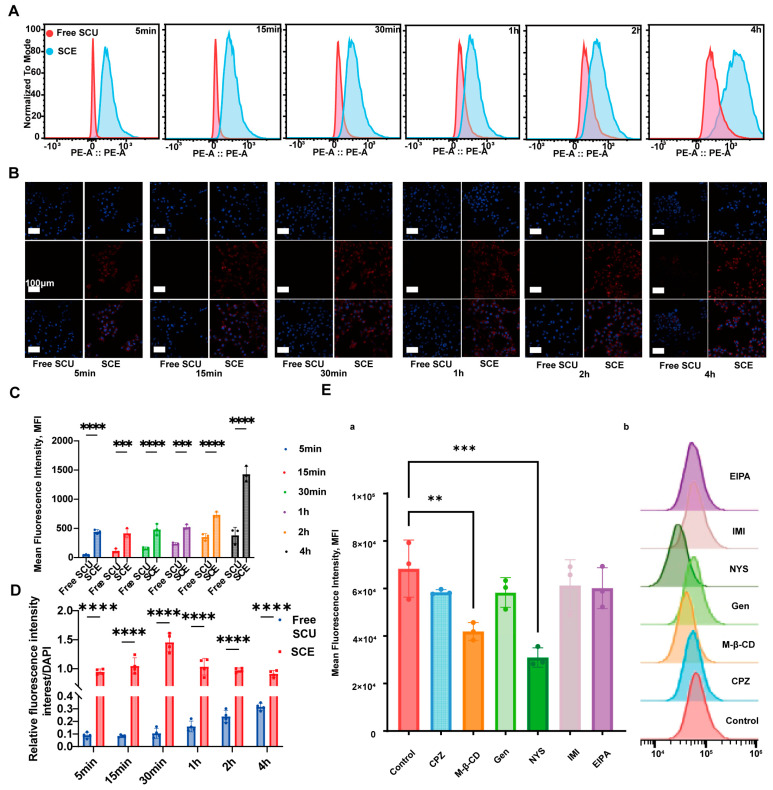
Enhanced cellular uptake of SCU by LX-2 hepatic stellate cells using the SCU nanoemulsion (SCE). (**A**) Flow cytometry representative histograms comparing Nile Red (NR) uptake over time. (**B**) Confocal microscopy images of LX-2 cells at 5 min, 1 h, and 4 h after treatment with free NR vs. NR-SCE. (**C**) Quantification of cellular NR uptake by flow cytometry (mean fluorescence intensity ± SD, n = 3). (**D**) Quantification of cellular NR uptake by Confocal microscopy. (Mean ± SD, n = 3). (**E**) Uptake of NR-SCE in LX-2 cells after pretreatment with endocytosis inhibitors. (**a**) Quantification of cellular NR uptake by flow cytometry (Mean ± SD, n = 3). (**b**) representative histograms. **** *p*< 0.0001, *** *p* < 0.001, ** *p* < 0.01.

**Figure 5 ijms-26-09746-f005:**
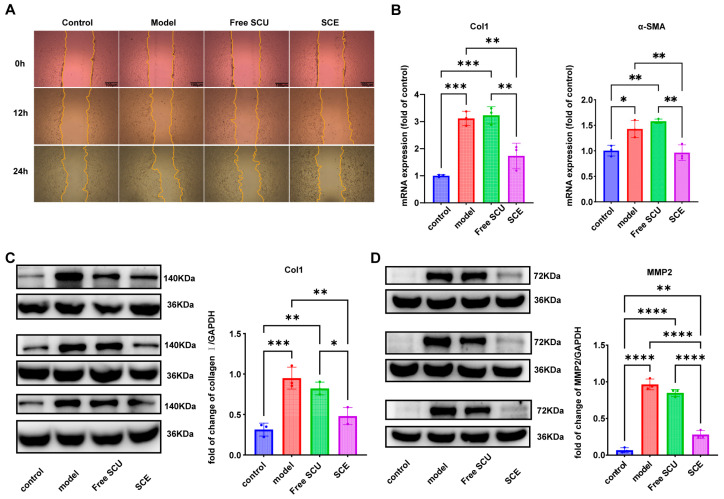
Anti-fibrotic effects of SCU (free vs. nanoemulsion) in TGF-β1-activated LX-2 hepatic stellate cells. (**A**) Wound healing (scratch) assay images at 0, 12, and 24 h for activated LX-2 cells under different treatments (no treatment, free SCU, SCE). (**B**) Relative mRNA expression of fibrosis-related genes (α-SMA and collagen I) in LX-2 cells after 12 h of treatment, as determined by qPCR (normalized to β-actin) (n = 3). (**C**,**D**) Western blot analysis of fibrotic proteins in LX-2 cells. (**C**) Collagen I and (**D**) MMP2 protein levels are shown (bands and quantification) (n = 3). **** *p* < 0.0001, *** *p* < 0.001, ** *p* < 0.01, * *p* < 0.05.

**Figure 6 ijms-26-09746-f006:**
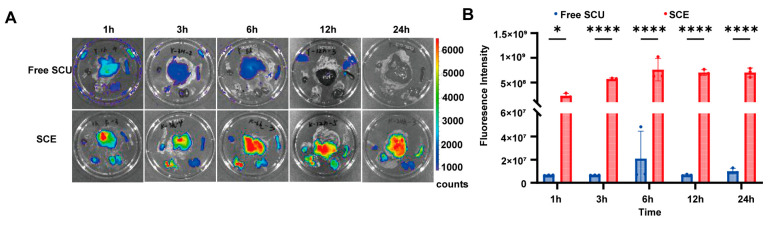
Enhanced hepatic accumulation and retention of scutellarin achieved by the nanoemulsion (SCE) in vivo. (**A**) Ex vivo fluorescence imaging of major organs (heart, liver, spleen, lung, and kidneys) collected from mice at 1, 3, 6, 12, and 24 h after oral administration of free DIR (a near-IR dye, surrogate for free SCU) vs. DIR-loaded SCE. (**B**) Quantification of liver fluorescence intensity over time (mean ± SD, n = 3). **** *p* < 0.0001, * *p* < 0.05.

**Figure 7 ijms-26-09746-f007:**
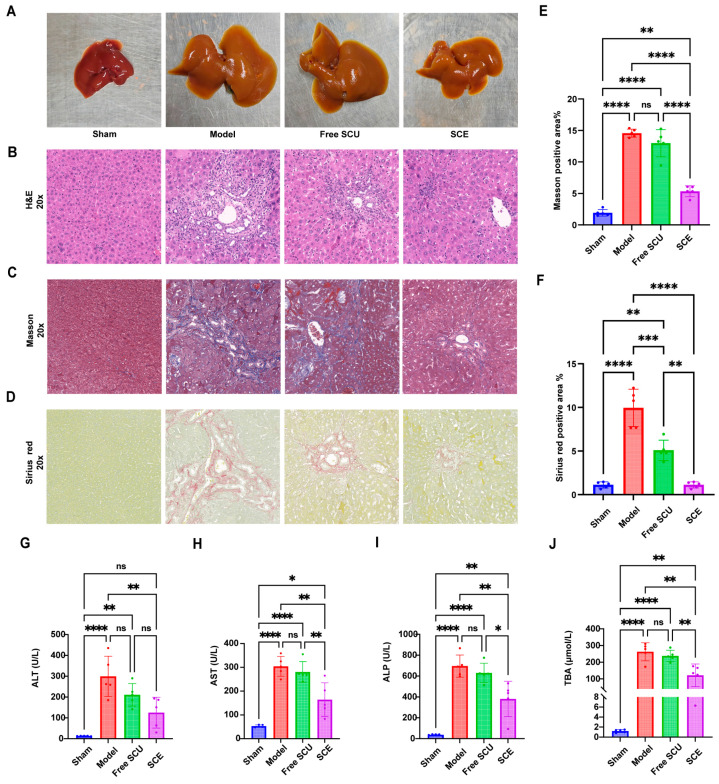
Therapeutic effects of SCE vs. free SCU in the BDL-induced mouse liver fibrosis model. (**A**) Representative gross morphology of livers from each group at day 14. (**B**) H&E-stained liver sections (20× magnification). (**C**) Masson’s trichrome staining for collagen (blue). (**D**) Sirius Red staining for collagen (red) corroborates Masson’s results. (**E**,**F**) Quantification of fibrotic area from Masson (**E**) and Sirius Red (**F**) stains (percentage of stained area ± SD, n = 5). (**G**–**J**) Serum levels of ALT, AST, ALP, and total bile acids (TBA) across different groups (mean ± SD, n = 5). Statistical analysis was conducted using the complete dataset. Q-values were calculated from *p*-values using the FDR method. **** *p* < 0.0001, *** *p* < 0.001, ** *p* < 0.01, * *p* < 0.05, ns, not significant (*p* ≥ 0.05).

**Figure 8 ijms-26-09746-f008:**
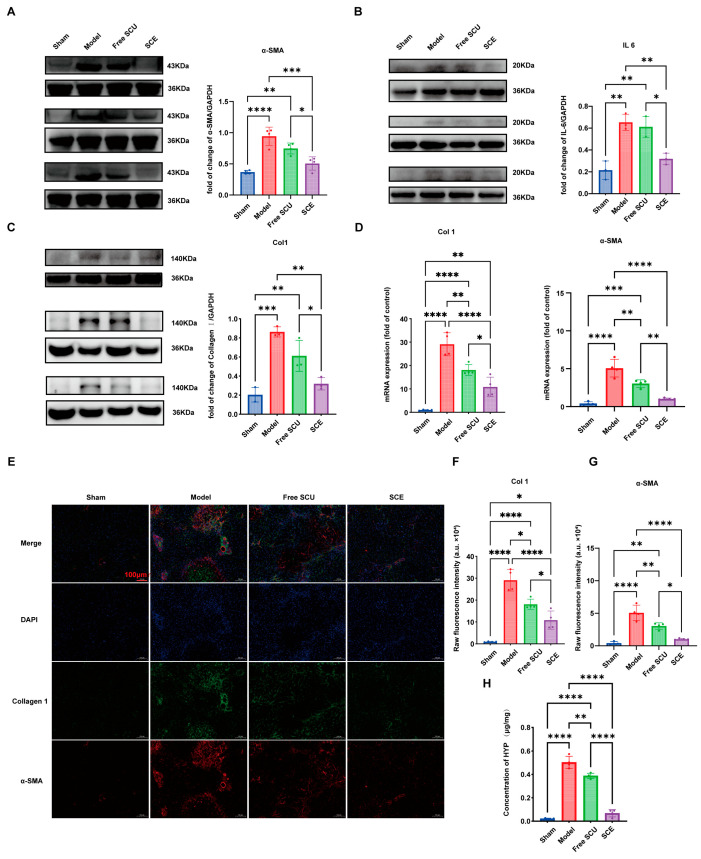
SCE treatment suppresses fibrogenic and inflammatory markers in fibrotic livers more effectively than free SCU. (**A**–**C**) Western blot analysis of liver tissue proteins (with GAPDH as loading control). (**A**) α-SMA, (**B**) IL-6, and (**C**) collagen I protein levels in Sham, BDL, free SCU, and SCE groups. (**D**) Quantitative PCR analysis of hepatic mRNA expression of α-SMA and collagen I. (**E**–**G**) Immunofluorescence staining of liver sections for collagen I (green) and α-SMA (red), with nuclei in blue (DAPI). (**H**) Hepatic hydroxyproline content (μg per gram of liver). Statistical analysis was conducted using the complete dataset. Q-values were calculated from *p*-values using the FDR method. **** *p* < 0.0001, *** *p* < 0.001, ** *p* < 0.01, * *p* < 0.05.

**Figure 9 ijms-26-09746-f009:**
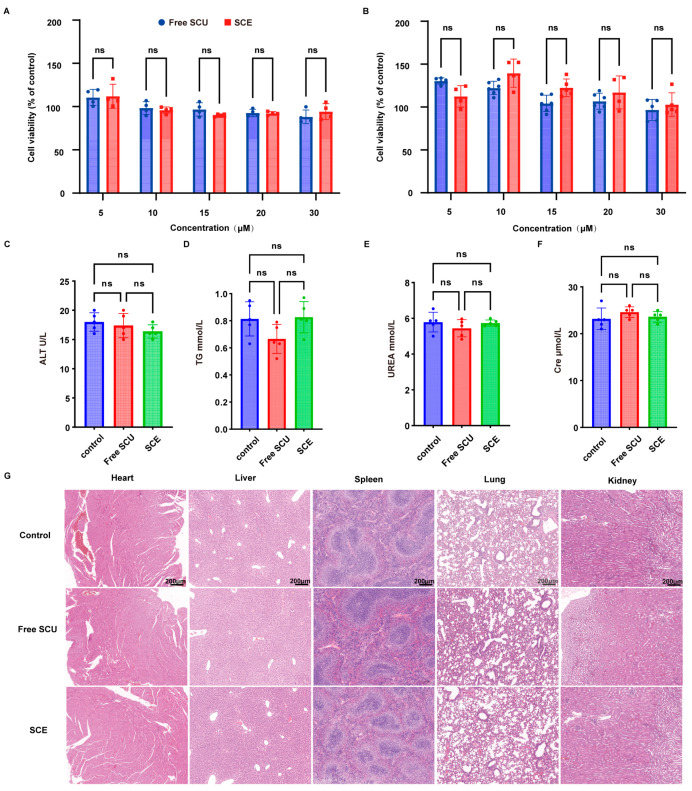
Biocompatibility assessment of the SCU nanoemulsion (SCE) in vitro and in vivo. (**A**,**B**) Cell viability of LO2 (normal liver) cells and LX-2 (HSC) cells after 24 h exposure to increasing concentrations of free SCU vs. SCE (SCU 5–30 µM), measured by CCK-8 assay. Data are presented as mean ± SD (n = 4). (**C**–**F**) Serum chemistry of mice following 14 days of treatment with saline (control), free SCU, or SCE (n = 5). (**G**) Histological examination (H&E staining) of major organs (heart, liver, spleen, lung, and kidney) from control, free SCU, and SCE-treated mice. ns, not significant (*p* ≥ 0.05).

**Table 1 ijms-26-09746-t001:** Physical stability of SCE at 4 °C.

Day	Mean PS ± SD	Mean PDI ± SD	Mean ZP ± SD	Mean Concentration ± SD
0	78.51 ± 2.45	0.23 ± 0.01	−3.6 ± 0.23	106.11 ± 3.49
1	81.93 ± 3.08	0.21 ± 0	−3.72 ± 0.35	98.39 ± 10.75
3	84.37 ± 3.2	0.20 ± 0.01	−3.86 ± 0.31	90.99 ± 18.58
5	86.62 ± 2.77	0.19 ± 0.01	−3.99 ± 0.38	105.22 ± 5.29
7	89.22 ± 2.94	0.19 ± 0.01	−3.77 ± 0.36	102.87 ± 5.56

## Data Availability

The datasets used and/or analyzed during the current study are available from the corresponding author upon reasonable request. The 16S rRNA data generated in this study will be deposited in the same repository and made publicly available upon acceptance of the manuscript for publication. Further details can be accessed at PRJNA1265637 and PRJNA1265571.

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
