# Peer review of "Liver-Targeted Scutellarin Nanoemulsion Alleviates Fibrosis with Ancillary Modulation of the Gut–Liver Microbiota"

_ijms, 2025, doi:10.3390/ijms26199746_

Round 1

Reviewer 1 Report

Comments and Suggestions for Authors

This manuscript presents an extensive study on scutellarin nanoemulsion (SCE) for liver fibrosis treatment. The work is interesting and the results are presented well. The overall merit is between average and high.

The biggest defect of the manuscript was that there were no independent cell experiments. If the describtion is ”Data are presented as mean ± SD (n=4 wells)” it means a single experiment. This might not be acceptable; the Editor has to make the decision on acceptance or rejection. The manuscript has so many good experiments in it that my personal opinion  is accepting after revisions.

If the manuscript is accepted, the claims should be less strong and the manuscript should take into notion and discuss the limited number of animals / independent replicates. For example, the claim that changes in microbiota may help the treatment effect of SCU was not directly tested. It would be better to say that microbiota changes are related to the effect (they cannot be claimed to be the main reason for it).

Minor remarks:

- Why there is only one compound tested? Please discuss why you did not include some other flavonoids in the experiment. Why scutellarin is better than for example silybin of quercetin?

- Is there a reference to the method of liver 16S sequencing? How was the contamination controlled?

- Instead of testing the stability of the SCE preparation on days 3 and 5, it would have been better to test longer time periods (1-6 months). Please include comment on the lack of knowledge on longer stability.

Author Response

Comments1:The biggest defect of the manuscript was that there were no independent cell experiments. If the description is “Data are presented as mean ± SD (n=4 wells)” it means a single experiment. This might not be acceptable. The manuscript has so many good experiments in it that my personal opinion is accepting after revisions.

Responds 1:We fully agree that this clarification is essential for accurate data interpretation, and we thank the reviewer for pointing this out. For cytotoxicity experiments, four independent experiments were performed on different days, and each experiment contained multiple technical replicates (n=4 wells). The statement “n=4 wells” in the original version may have inadvertently suggested a single experiment, which was misleading. We have now corrected this throughout the text and figure legends to clearly distinguish biological replicates (independent experiments) from technical replicates (multiple wells within one experiment).

In the revised version, we have clarified the experimental design in the Methods section on page 6, line 209-213. We also clarified this in figure legend on page 28, line 976.

Comment 2: If the manuscript is accepted, the claims should be less strong and the manuscript should take into notion and discuss the limited number of animals / independent replicates. For example, the claim that changes in microbiota may help the treatment effect of SCU was not directly tested.

Respond 2: We thank the reviewer for this crucial feedback regarding the interpretation of our data, particularly concerning the microbiome. We have softened the claims throughout the manuscript.

In the revised Abstract (on page 1, line 24-25, 29), Results (on page 15, line 570-572), Figure legend (on page 14, line 554) and Discussion (on page 29, line 1015-1019 and line 1022-1025 and 1027-1029), we now describe microbiota modulation as “associated with” or “potentially supportive of” the anti-fibrotic effects, rather than suggesting it is the primary causal mechanism. Additionally, we have added a limitation statement in the Discussion (on page 29, line 1026), acknowledging that the animal number is limited and that further studies with larger cohorts and causal validation (e.g., fecal microbiota transplantation) are warranted.

Comment 3: Why is there only one compound tested? Please discuss why you did not include some other flavonoids in the experiment. Why scutellarin is better than for example silybin or quercetin?

Response 3: We thank the reviewer for this thoughtful comment. Our study focused on scutellarin (SCU) for several reasons: (i) it is a well-characterized flavonoid with documented anti-fibrotic (references 16, 17) and microbiota-modulating activities (references 5, 6); (ii) it has inherently poor solubility and oral bioavailability, which makes it an ideal candidate to demonstrate the advantages of our nanoemulsion delivery strategy; and (iii) it has been widely used in traditional Chinese medicine with an established safety profile, thereby supporting its translational relevance. We fully agree that other flavonoids such as silybin and quercetin are also promising anti-fibrotic candidates.

To address this point, we have added a paragraph in the Discussion section (on page 29, line 990-995) acknowledging their therapeutic potential and clarifying that the nanoemulsion approach could in principle be extended to these and other natural flavonoids in future work.

Reference 16: DOI: 10.1111/j.1476-5381.2010.01070.x

Reference 17: DOI: 10.1038/s41419-020-03178-2

Comment 4: Is there a reference to the method of liver 16S sequencing? How was the contamination controlled?

Response 4: We appreciate this request for clarification. Specifically: Liver samples were collected under aseptic conditions, and DNA extraction/PCR reagents were handled in a sterile environment. While we did not include dedicated blank extraction controls, all experimental groups were processed in parallel under identical conditions, and interpretation was based on relative differences between treatment and control groups rather than absolute abundance.

In the revised Methods (on page 9, line 399), we have now added references (20, 21) for the 16S rRNA sequencing workflow (QIIME2 pipeline, SILVA database) and clarified our procedures for contamination control. In addition, we have emphasized in the Discussion(on page 29, line 1015-1018, 1022-1025, 1029-1031) that liver microbiota data should be considered exploratory, given the low microbial biomass in liver tissue and the ongoing debate regarding potential contamination in extraintestinal microbiome studies.

Reference 20: DOI: 10.1172/JCI151725

Reference 21: DOI: 10.1016/j.jhepr.2021.100299

Comment 5: Instead of testing the stability of the SCE preparation on days 3 and 5, it would have been better to test longer time periods (1-6 months). Please include comment on the lack of knowledge on longer stability.

Response 5: We agree with the reviewer that long-term stability data is important for translational potential. Our 7-day study was designed to ensure formulation stability during the in vitro and in vivo experimental period.

We have now added a statement in the Results (on page 17, line 646-649) and Discussion (on page31, line 1100-1104) acknowledging that long-term stability (1-6 months) was not assessed here, and that further optimization and extended stability testing will be necessary before clinical translation.

Reviewer 2 Report

Comments and Suggestions for Authors

The authors investigated the therapeutic potential of a Scutellarin-loaded nanoemulsion in a bile duct ligation mouse model of liver fibrosis, focusing on its antifibrotic effects and its ability to modulate both gut and liver microbiota.

The manuscript overstates its degree of novelty. While the dual targeting of hepatic fibrogenesis and gut microbiota using a phytochemical nanoformulation is interesting, similar strategies have been reported previously. The findings primarily confirm known antifibrotic and microbiota-modulating effects of Scutellarin rather than providing fundamentally new mechanistic insights. The translational impact therefore appears limited.

The methodological design of the study presents several shortcomings. The stability of the nanoemulsion was tested for only 7 days at 4°C, a duration too short to substantiate claims of practical applicability. Although the bile duct ligation (BDL) model is suitable, the authors evaluated only a single dose of the Scutellarin nanoemulsion, and the absence of a dose-response assessment limits the strength of the conclusions. Furthermore, the microbiome analysis is exploratory in nature and prone to contamination due to low biomass samples, an issue the authors themselves acknowledge, which diminishes confidence in the liver microbiota findings. The lack of sample size justification further undermines reliability, with certain experiments, particularly those measuring serum markers (n=3), being statistically underpowered. In addition, the in vitro experiments were not adequately controlled, as they did not include a vehicle-only nanoemulsion group, which restricts the validity of the observed effects.

The use of one-way ANOVA with Tukey’s post-hoc test is reported, but multiple outcomes were assessed without correction for multiple testing, increasing the risk of false positives. Data presentation as mean ± SD may not be appropriate for non-normally distributed variables, such as microbiota abundances. Effect sizes and confidence intervals are not provided, limiting the interpretability of the results.

The authors claim fibrosis regression; however, the data demonstrate attenuation rather than true reversal. They present several microbiota shifts as meaningful despite lacking statistical significance. Their assertion of a “slight cytoprotective effect” is speculative and not supported by mechanistic evidence. Furthermore, the authors place important quantitative findings in the supplementary material, which reduces transparency and limits accessibility.

Several figures are overcrowded with panels, which compromises readability. The figure legends include interpretative statements rather than neutral descriptions. Moreover, the fragmentation of results between main and supplementary figures reduces clarity and hampers the coherence of data presentation.

The discussion primarily reiterates results rather than providing a critical evaluation. While limitations such as microbiome contamination risk and insufficient sample size are mentioned, they are downplayed rather than thoroughly examined. Potential confounding effects of nanoemulsion lipids are not discussed in sufficient depth. The conclusions are disproportionately strong relative to the data; phrases such as “safe and effective nanomedicine” are premature in the absence of long-term toxicity, pharmacokinetic, and comprehensive dose-response studies. Finally, the proposed causal link between microbiota modulation and antifibrotic effects remains correlative and unsubstantiated.

Comments on the Quality of English Language

A thorough language revision is strongly recommended. Several issues affect readability:

- typographical errors (e.g., double periods at lines 326 and 370).

- word duplication (“LX-2 cells LX-2 cells”, line 670).

- awkward phrasing (e.g., line 1011: “for SCE in LX-2 suggests”, should be “SCE in LX-2 cells suggests”).

- the overuse of passive voice reduces clarity throughout.

English including grammar, style and syntax, should be improved through the professional help from English Editing Company for Scientific Writings.

Author Response

Comment 1: The manuscript overstates its degree of novelty. While the dual targeting of hepatic fibrogenesis and gut microbiota using a phytochemical nanoformulation is interesting, similar strategies have been reported previously. The findings primarily confirm known antifibrotic and microbiota-modulating effects of Scutellarin rather than providing fundamentally new mechanistic insights. The translational impact therefore appears limited.

Respond 1: We sincerely thank the reviewer for this valuable comment. We have carefully revised the manuscript to more accurately define the novelty of our study. We agree that the antifibrotic and microbiota-modulating effects of scutellarin (SCU) have been previously reported, and our intention was not to claim discovery of new biological properties. Rather, the novelty of our work lies in the engineering of a nanoemulsion delivery system that addresses SCU’s critical limitations of solubility and bioavailability, thereby enabling effective hepatic targeting and amplifying its therapeutic potential in liver fibrosis.Our study provides integrated evidence that improved hepatic accumulation of SCU can be coupled with ancillary modulation of the gut-liver microbiota. We acknowledge that this evidence is associative rather than causal.

We point out our novelty on page 29-30, line 1037-1041 and we have revised the Discussion (on page 29, line 1012-1020).

Comment 2: The methodological design of the study presents several shortcomings. The stability of the nanoemulsion was tested for only 7 days at 4°C, a duration too short to substantiate claims of practical applicability.

Respend 2: We agree with the reviewer that long-term stability data is important for translational potential. Our 7-day study was designed to ensure formulation stability during the in vitro and in vivo experimental period.

We have now added a statement in the Results (on page 17, line 646-649) and Discussion (on page31, line 1100-1104) acknowledging that long-term stability (1-6 months) was not assessed here, and that further optimization and extended stability testing will be necessary before clinical translation.

Comment 3: Although the bile duct ligation (BDL) model is suitable, the authors evaluated only a single dose of the Scutellarin nanoemulsion, and the absence of a dose-response assessment limits the strength of the conclusions.

Respend 3: We acknowledge this limitation. Due to ethical considerations of animal use, we employed a single dose as a proof-of-concept study. The selected dose (10 mg/kg) was based on prior literature with free SCU and our preliminary studies aimed at achieving a therapeutic effect (references 43, 44).

We have added this to the Discussion (on page 31, line 1099-1102) and highlighted the need for future dose–response experiments.

Reference 43: DOI: 10.1177/09603271211045948

Reference 44: DOI: 10.1248/bpb.b23-00390

Comment 4: Furthermore, the microbiome analysis is exploratory in nature and prone to contamination due to low biomass samples, an issue the authors themselves acknowledge, which diminishes confidence in the liver microbiota findings

Respend 4: Thank you for this critical and insightful comment. You raise a fundamental point regarding the challenges of low-biomass microbiome studies, and we fully agree that the potential for contamination is a significant concern that may undermine confidence in the findings.

While we did not apply advanced bioinformatic decontamination tools (e.g., decontam), we implemented stringent laboratory measures to minimize contamination at its source: all sample processing and DNA extraction were performed under a Class II biosafety cabinet using sterile, DNA-free reagents and consumables. Importantly, our analysis was conservative, focusing on relative differences between study groups rather than asserting the absolute presence of specific taxa. The rationale is that although background contamination may exist, if it is evenly distributed, differential signals between groups may still reflect underlying biological variation.

In the revised manuscript, we have added explicit statements in the Discussion (on page 29, line 1015-1031) to acknowledge this limitation. We now emphasize that the liver microbiota data are exploratory in nature, lack formal bioinformatic decontamination, and should therefore be interpreted with caution as hypothesis-generating rather than definitive. Correspondingly, we have tempered our conclusions throughout the manuscript to ensure they align with this cautious interpretation.

We sincerely appreciate the reviewer’s comment, which has improved the rigor, balance, and transparency of our work.

Comment 5: The lack of sample size justification further undermines reliability, with certain experiments, particularly those measuring serum markers (n=3), being statistically underpowered.

Respend 5: Thank you for pointing out this important issue. We apologize for the confusion caused by the way sample sizes were reported. In fact, each experimental group originally included 5-6 animals, but due to incomplete data collection for certain serum markers, only three values were initially presented. We have now re-analyzed the dataset using the full sample size (n=5 per group), and updated the figures and statistical analyses accordingly (Figure 7). The revised results are statistically sound and further support the conclusions.

We have also clarified the sample size information in the Methods (on page 8, line 335-339) and figure legends (on page 24, line 867) to avoid ambiguity.

Comment 6 In addition, the in vitro experiments were not adequately controlled, as they did not include a vehicle-only nanoemulsion group, which restricts the validity of the observed effects.

Respend 6: We thank the reviewer for this important comment. We agree that including a blank nanoemulsion vehicle control would have strengthened the rigor of the in vitro experiments. This control was not incorporated in the present study, which means that potential effects attributable to the lipid components of the vehicle cannot be entirely excluded.

In the revised manuscript, we now explicitly acknowledge this limitation in the Discussion (on page 31, line 1107-1114) and emphasize that future studies will incorporate vehicle-only controls to fully validate the specificity of the observed effects.

Comment 7: The use of one-way ANOVA with Tukey’s post-hoc test is reported, but multiple outcomes were assessed without correction for multiple testing, increasing the risk of false positives. Data presentation as mean ± SD may not be appropriate for non-normally distributed variables, such as microbiota abundances. Effect sizes and confidence intervals are not provided, limiting the interpretability of the results.

Respend 7: We thank the reviewer for these valuable suggestions, which have helped us to improve the statistical rigor of our study.

In the revised manuscript:

1.We have applied the Benjamini-Hochberg false discovery rate (FDR) correction to all multiple comparisons, and results are now interpreted based on FDR-adjusted q-values. (Figure 7,8 and their legnd)

2.For data not following a normal distribution (e.g., microbiota abundances), values are now presented as median with interquartile range (IQR) rather than mean ± SD. We have also added a statement in the Methods (on page 10, line 442-446) and Discussion (on page 29, line 1029-1031) acknowledging the distributional characteristics of these data and the need for cautious interpretation.

We have added a paragraph in the Discussion (on page 31, line 1116-1117) explicitly acknowledging that effect sizes and confidence intervals were not provided, which limits the interpretability of some results. We agree that this is an important issue that should be addressed in future studies, and we have tempered our conclusions accordingly.

Comment 8: The authors claim fibrosis regression; however, the data demonstrate attenuation rather than true reversal. They present several microbiota shifts as meaningful despite lacking statistical significance. Their assertion of a “slight cytoprotective effect” is speculative and not supported by mechanistic evidence. Furthermore, the authors place important quantitative findings in the supplementary material, which reduces transparency and limits accessibility.

Respend 8: We thank the reviewer for this constructive comment. We agree that our data support attenuation of fibrosis rather than true regression, and we have revised the text accordingly to use more accurate wording (e.g., “attenuation” or “amelioration”). Regarding the microbiota results, we have clarified in the revised manuscript that some of the observed shifts represent non-significant trends and should be interpreted cautiously. We have also tempered our language in the Discussion to emphasize that these findings are exploratory. Finally, we acknowledge that our statement on a “slight cytoprotective effect” was speculative and not directly supported by mechanistic data; this has been removed to avoid overinterpretation.

Comment 9: ...the authors place important quantitative findings in the supplementary material...

Respend 9: We thank the reviewer for raising this point. We carefully re-examined and adjusted our supplementary material and can confirm that all major quantitative results are presented in the main text and figures now. The supplementary material only contains supporting information (e.g., additional methodological details and representative images/figures) intended to complement, but not replace, the main findings.

Comment 10: Several figures are overcrowded with panels, which compromises readability. The figure legends include interpretative statements rather than neutral descriptions. Moreover, the fragmentation of results between main and supplementary figures reduces clarity and hampers the coherence of data presentation.

Respend 10: We thank the reviewer for raising this point. All figure legends have been revised to provide concise and neutral descriptions of the content, avoiding interpretative statements, which are now moved into the Results or Discussion text. To enhance clarity and coherence, we have consolidated the presentation of results: key quantitative findings are retained in the main figures, while supplementary figures now contain only representative or supporting information

We believe these changes substantially improve the readability and coherence of our data presentation.

Comment 14: The conclusions are disproportionately strong... phrases such as “safe and effective nanomedicine” are premature...

Respend 14: We have significantly toned down the conclusion. We now describe SCE as a "promising therapeutic strategy" or "a promising formulation" based on our pre-clinical proof-of-concept data, explicitly stating that further development is required before clinical application can be considered.

Round 2

Reviewer 2 Report

Comments and Suggestions for Authors

The authors mostly responded to the comments and suggestions and the manuscript was revised accordingly. I consider it could be accepted for publication in this journal.

Comments on the Quality of English Language

I propose to have the manuscript checked by a native English speaking person.

Author Response

Comments 1: The authors mostly responded to the comments and suggestions and the manuscript was revised accordingly. I consider it could be accepted for publication in this journal.

Respond 1: We would like to sincerely thank you for your valuable comments and suggestions on our manuscript. Your professional feedback has been very meaningful to our research, and we will pay closer attention to these important points in our future work. We are truly grateful for your thoughtful review and constructive guidance, which have helped us to strengthen the quality of our work.

Comments 2: I propose to have the manuscript checked by a native English speaking person.

Respond 2: Regarding the English language, we carefully revised the manuscript and also had it polished by a professional editing service to further improve clarity and readability, making the text more accessible to readers.
